# MATCHING MULTIPLE EXPERTS: ON THE EXPLOITABILITY OF MULTI-AGENT IMITATION LEARNING

**Antoine Bergerault**[*]
EPFL, UC Berkeley

**Volkan Cevher**
EPFL

**Negar Mehr**
UC Berkeley

## ABSTRACT

Multi-agent imitation learning (MA-IL) aims to learn optimal policies from expert demonstrations of interactions in multi-agent interactive domains. Despite existing guarantees on the performance of the resulting learned policies, characterizations of how far the learned polices are from a Nash equilibrium are missing for offline MA-IL. In this paper, we demonstrate impossibility and hardness results of learning low-exploitable policies in general $n$-player Markov Games. We do so by providing examples where even exact measure matching fails, and demonstrating a new hardness result on characterizing the Nash gap given a fixed measure matching error. We then show how these challenges can be overcome using strategic dominance assumptions on the expert equilibrium. Specifically, for the case of dominant strategy expert equilibria, assuming Behavioral Cloning error $\epsilon_{\text{BC}}$, this provides a Nash imitation gap of $\mathcal{O}\left(n\epsilon_{\text{BC}}/(1-\gamma)^2\right)$ for a discount factor $\gamma$. We generalize this result with a new notion of best-response continuity, and argue that this is implicitly encouraged by standard regularization techniques.

## 1 INTRODUCTION

Learning from expert demonstrations via imitation learning (IL) has recently seen growing adoption in the Machine Learning and Robotics communities (Finn et al., 2016b; Shih et al., 2022; Pearce et al., 2023; Yang et al., 2023). Given a demonstration dataset, IL is traditionally done by either regressing a policy (Behavioral Cloning (Pomerleau, 1991)), fitting a plausible reward function and extracting a policy via Reinforcement Learning (Inverse Reinforcement Learning (Ng et al., 2000; Abbeel & Ng, 2004)), or implicitly matching expert occupancy measures (Finn et al., 2016a; Ho & Ermon, 2016). Crucially, imitation learning bypasses the need of designing a reward function, a common limitation for Reinforcement Learning in practice, that often requires domain expertise or extensive iterative refinements. Instead, it directly leverages demonstrations from optimal agents. This advantage becomes even more compelling when learning tasks requiring collaboration or competition between multiple agents, where reward assignment constitutes an extra ambiguity (Sunehag et al., 2017; Nagpal et al., 2025; Choi et al., 2025).

While many works successfully tackle single agent IL (SA-IL, Ho & Ermon (2016); Ng et al. (2000); Ross & Bagnell (2010)), their extensions to multi-agent settings (Song et al., 2018; Zhan et al., 2018; Mehr et al., 2023) inherit fundamental limitations. In particular, they produce imitation policies that can be exploited by unilaterally deviating strategic agents (Tang et al., 2024; Freihaut et al., 2025).

In this work, we study the question of learning a Nash equilibrium using demonstrations from an expert Nash equilibrium, given fixed imitation errors commonly measured in practice. More precisely, assuming a known behavioral cloning or measure matching error, we measure the exploitability of the learned policy as its distance to a Nash equilibrium. We characterize situations where we can derive both *consistent* and *tractable* bounds on the Nash gap (see Section 3 for a formal definition), where

- A *consistent* bound vanishes with the imitation error; making the imitation error an intuitive proxy of the exploitability by guaranteeing a pure Nash equilibrium from no imitation error.
- A *tractable* bound is efficient to compute based on the game assumptions. It can be computed in polynomial time to measure exploitability during IL training.

---

[*]The work was done when the first author was an EPFL master's student visiting UC Berkeley. Now at the University of Zurich. Correspondence to antoine.bergerault@uzh.ch.

Intuitively, a *consistent* bound ensures that a Nash equilibrium is learned from an imitation error of zero. Consistency can be implicitly assumed by SA-IL extensions to multi-agent domains, but *we show that this is a strong assumption that does not hold in general games*.

Specifically, we prove how not assuming full-state support of the expert or only matching its state distribution can lead to bound inconsistencies. Then, we present continuity conditions under which both *consistent* and *tractable* upper bounds on the Nash gap can be computed. In summary, we make the following contributions:

- In Section 4, we show the impossibility of deriving *consistent* Nash gap bounds in general Markov Games. We provide concrete examples where even exactly matching expert occupancy measures can result in highly exploitable policies.

- We further demonstrate in Section 5 the impossibility of deriving tight *tractable* exploitability lower bounds in general games, even if we know both the rewards and transition dynamics.

- Finally, Section 6 presents a new notion of best-response continuity not observed in SA-IL and shows how assumptions on this continuity property can be used to construct *tractable* upper bounds. As a special case, we prove that a "good" approximate Nash equilibrium can be learned from Behavioral Cloning with a dominant strategy expert.

To keep the arguments straightforward, we present our results for infinite-horizon games in the main document. These results also extend to finite-horizon games, as demonstrated in Appendix F.

## 2 PREVIOUS WORK

**Single-agent Imitation Learning.** Given a dataset of demonstrations produced by an expert, SA-IL aims to extract a near-optimal policy from the data. The expert is considered optimal in maximizing a reward function over time, as in the reinforcement learning framework. Without requiring access to the environment or expert oracles, imitation learning is done through Behavioral Cloning (BC, Pomerleau (1991)), Inverse Reinforcement Learning (IRL, Ng et al. (2000); Abbeel & Ng (2004)) or Adversarial Imitation Learning (Ho & Ermon, 2016). These methods essentially fit one of: the expert policy function, the reward function, or the expert occupancy measures (Finn et al., 2016a; Ho & Ermon, 2016). In the single-agent setting, performance of such approaches are well-understood, measured by the sub-optimality gap of the learned policy with respect to the expert (Ross et al., 2011; Foster et al., 2024).

**Multi-agent Imitation Learning.** A growing body of work focuses on imitation learning in multi-agent domains (Song et al., 2018; Lin et al., 2018; Wang et al., 2021; Shih et al., 2022; Mehr et al., 2023), with applications such as autonomous driving (Bhattacharyya et al., 2018) or robotic interactions (Bogert & Doshi, 2018; Chandra et al., 2025). MA-IRL inherits the ambiguity of reward design from the reinforcement learning framework (Sunehag et al., 2017; Freihaut & Ramponi, 2025; Ward et al., 2025). More simple methods (BC, Adversarial IL) are therefore tempting and have been extended from their single-agent counterpart (Song et al., 2018; Zhan et al., 2018). However, they do not carry guarantees on the extracted policy in terms of robustness to the presence of strategic interactions.

**Theoretical Barriers for MA-IL.** Indeed, previous work showed that BC and GAIL policies are exploitable in general games (Cui & Du, 2022; Freihaut et al., 2025; Tang et al., 2024). Among those works, Freihaut et al. (2025) introduces the first regret upper bound for multi-agent behavioral cloning, bounding the sub-optimality of the imitation policy from any unilateral deviations. They rely on a new concentrability coefficient related to broader concentrability assumptions (Cui & Du, 2022; Yin et al., 2021; Cai et al., 2023). This coefficient is *intractable* in general games and can become unbounded, making their bound both *inconsistent* and hard to use in practice.

There remains a gap in the literature in identifying the phenomena behind impossibility results from prior work, and conditions to make offline MA-IL well-behaved are still unclear. We reduce this gap by characterizing such issues and deriving conditions for *consistent* and *tractable* Nash gap bounds.

## 3 PRELIMINARIES

### 3.1 MARKOV GAMES

We use the tuple $G = (\mathcal{S}, \mathcal{A}, P, \{r_i\}_{i=1}^n, \nu_0, \gamma)$ to define an $n$-player Markov Game. Players in $[n] := \{1, \ldots, n\}$ take joint actions in $\mathcal{A} = \mathcal{A}_1 \times \cdots \times \mathcal{A}_n$ while navigating a shared state space $\mathcal{S}$. The dynamics of the system are described by the transition function $P : \mathcal{S} \times \mathcal{A} \to \Delta_{\mathcal{S}}$ and the initial state distribution $\nu_0 \in \Delta_{\mathcal{S}}$, where $\Delta_U$ denotes the probability simplex over a space $U$. All players simultaneously take actions by sampling from their individual policy $\pi_i : \mathcal{S} \to \Delta_{\mathcal{A}_i}$. The resulting joint policy is $\pi := \pi_1 \times \cdots \times \pi_n$, also denoted by $\pi_i \times \pi_{-i}$ where $\pi_{-i}$ represents the joint policy of all players but $i$. Lastly, we define the reward function of player $i \in [n]$ as $r_i : \mathcal{S} \times \mathcal{A} \to [-1, 1]$, and define a discount factor $\gamma \in [0, 1)$. We will use $\Pi = \Pi_1 \times \cdots \times \Pi_n$ to denote the set of joint policies $\pi$, with $\pi_i \in \Pi_i$ for each $i \in [n]$.

A trajectory $\{(s_t, a_t)\}_{t \geq 0}$ of the policy $\pi$ starts in an initial state $s_0 \sim \nu_0$ and successively samples joint actions $a_t \sim \pi(\cdot|s_t)$ and future states $s_{t+1} \sim P(\cdot|s_t, a_t)$ for every $t \geq 0$. This procedure induces the following state-only and state-action occupancy measures for every state $s \in \mathcal{S}$, respectively:

$$\mu_\pi(s) := (1 - \gamma) \sum_{t=0}^{\infty} \gamma^t \mathbb{P}(s_t = s), \qquad \rho_\pi(s, a) := \mu_\pi(s) \pi(a|s),$$

where $\mathbb{P}(s_t = s)$ is the probability of reaching state $s$ after rolling out the policy $\pi$ for $t$ steps. Intuitively, $\mu_\pi(s)$ is the discounted visitation frequency of state $s$ after infinitely many steps. Similarly, $\rho_\pi(s, a)$ is the discounted frequency of the state-action pair $(s, a)$. This allows us to define for every state $s \in \mathcal{S}$, the state-value functions as the expected discounted cumulative rewards of the players:

$$V_i^\pi(s) = \frac{1}{1 - \gamma} \cdot \mathbb{E}_{(s,a) \sim \rho_\pi}[r_i(s, a)] \quad \forall i \in [n],$$

where $\mathbb{E}_{(s,a) \sim \rho_\pi}[\cdot]$ samples state-action pairs from the density function $\rho_\pi$. By extension, we also define $V_i^\pi(\nu) = \mathbb{E}_{s \sim \nu}[V_i^\pi(s)]$ for any state distribution $\nu \in \Delta_{\mathcal{S}}$.

Markov Games extend Markov Decision Processes (MDPs, when $n = 1$ (Puterman, 2014)) to multi-agent games, where each agent has its own reward function. While in MDPs we consider a policy to be optimal if it maximizes $V^\pi(\nu_0)$[1], the distinct individual rewards of a Markov Game necessitate the introduction of the solution concept of a game-theoretic equilibrium to model the outcome of interactions.

### 3.2 MEASURING OPTIMALITY IN GAMES

Fixing other players' policies as $\pi_{-i}$, the performance of a player $i$ in terms of average (discounted) rewards is measured with $V_i^\pi(\nu_0)$. Therefore, we can denote the set of optimal policies for player $i$ as the optimal policies in the MDP induced by $\pi_{-i}$ using the concept of best-response mapping.

**Definition 1** (Best-response mapping). *For an agent $i \in [n]$, the best-response mapping to $\pi_{-i}$ is defined as*

$$\mathrm{BR}_i(\pi_{-i}) := \arg\max_{\pi_i'} V_i^{\pi_i', \pi_{-i}}(\nu_0).$$

Intuitively, when playing a best-response $\pi_i^* \in \mathrm{BR}_i(\pi_{-i})$, player $i$ cannot improve by unilaterally deviating from $\pi_i^*$. From this definition, we define a Nash equilibrium as a combination of independent policies where no player would be better off by unilaterally deviating from their equilibrium policies.

**Definition 2** (Nash equilibrium). *A policy $\pi$ is a Nash equilibrium of the game if $\pi$ is a product policy and each individual policy is a best-response to the other policies, i.e.*

$$\pi_i \in \mathrm{BR}_i(\pi_{-i}) \quad \forall i \in [n].$$

As is common in multi-agent games, we use Nash equilibria as solution concepts to model interaction outcomes throughout this paper. Specifically, our goal is to learn (approximate) Nash equilibria[2] from a dataset of trajectories sampled from a Nash equilibrium policy $\pi^E$ termed the *expert policy*.

---

[1] Or $V_1^\pi(\nu_0)$ using the notation above

[2] An (approximate) $\epsilon$-Nash equilibrium $\pi^E$ is a product policy where $V_i^{\pi_i, \pi_{-i}^E}(\nu_0) - V_i^{\pi^E}(\nu_0) \leq \epsilon$ for all $i \in [n], \pi_i \in \Pi_i$.

### 3.3 Offline Imitation learning

Given a dataset of finite-length trajectories[3] produced by rolling-out $\pi^E$ in a given Markov Game $G$, an imitation learning procedure aims to recover a "good" joint policy $\pi$ without access to the environment.

Following the above discussion, we measure the performance of $\pi$ with the following metric.

**Definition 3** (Value gap). *Given an expert policy $\pi^E$ of G, the Value gap of a policy $\pi$ is:*

$$\text{ValueGap}(\pi) := \max_i \left( V_i^{\pi^E}(\nu_0) - V_i^{\pi}(\nu_0) \right).$$

This is essentially the maximum sub-optimality gap among the agents for playing $\pi$ instead of the optimal expert $\pi^E$. This value gap is the standard optimality metric used in single-agent MDP settings (Ross et al., 2011; Foster et al., 2024).

In the multi-agent case, however, the performance of individual players is usually not a sufficient guarantee as we use the imitation policy in an environment with strategic agents. Hence, we need to measure the impact of individual policies deviating from the learned joint behavior $\pi$. We will therefore evaluate the exploitability of $\pi$, as a measure of its gap to a Nash equilibrium.

**Definition 4** (Nash gap, see Ramponi et al. (2023)). *We define the Nash (imitation) gap of a product policy $\pi$ of the game G as*

$$\text{NashGap}(\pi) := \max_i \left( V_i^{\pi_i^*, \pi_{-i}}(\nu_0) - V_i^{\pi}(\nu_0) \right) \tag{1}$$

*with any $\pi_i^* \in \text{BR}_i(\pi_{-i})$.*

The Nash gap is a notion of maximal regret (Tang et al., 2024) for the individual players. It directly links to Nash equilibria as an $\epsilon$-Nash equilibrium is any product policy $\pi_\epsilon \in \Pi$ satisfying $\text{NashGap}(\pi_\epsilon) \leq \epsilon$. Note that to ensure the imitation policy $\pi$ is a product policy, it suffices to learn all individual policies $\pi_i$ independently, for example through behavioral cloning.

In the general case, both $\text{ValueGap}(\pi)$ and $\text{NashGap}(\pi)$ are upper bounded by $\frac{2}{1-\gamma}$ as differences of cumulative normalized rewards. The goal of an imitation learning procedure is to leverage the expert data in order to minimize these gaps.

Regressing on the training data, BC and Adversarial IL bring one of the following error assumptions: a **BC Error** from matching the empirical distribution of the individual agents, or a **Measure Matching Error** measuring a discrepancy in occupancy measures (state-only or state-action). They are respectively defined as:

$$\epsilon_{\text{BC}} := \max_i \mathbb{E}_{s \sim \mu_{\pi^E}} \left[ \left\| \pi_i(\cdot|s) - \pi_i^E(\cdot|s) \right\|_1 \right], \tag{2}$$

$$\epsilon_\mu := \| \mu_\pi - \mu_{\pi^E} \|_1, \quad \text{and} \quad \epsilon_\rho := \| \rho_\pi - \rho_{\pi^E} \|_1,$$

where $\mathbb{E}_{s \sim \mu_{\pi^E}}[\cdot]$ is the expectation with respect to the state-occupancy induced by $\pi^E$.

While general *consistent* and *tractable* upper bounds are known on the Value gap assuming either $\epsilon_{\text{BC}}$ or $\epsilon_\rho$[4], deriving similar bounds for the Nash gap remains an open problem. Under the assumption of a fixed imitation error, we therefore lack an understanding of the distance of $\pi$ to a Nash equilibrium.

More information about the connection between adversarial imitation learning and occupancy measure matching can be found in Appendix A.

## 4 Impossibility results for exact measure matching

As a first step to understand the difficulty of extracting Nash equilibria from expert demonstrations, we focus in this section on the idealized case of exact occupancy measure matching. This is a crucial step for determining when a bound can be *consistent*, while the assumption is relaxed in later sections. Specifically, this section addresses the following question:

---

[3]While we consider stationary policies maximizing cumulated rewards over an infinite horizon, IL usually assumes a set of $N$ trajectories $\{\tau_k\}_{k=1}^N$ of length $|\tau_k| \sim \text{Geometric}(1-\gamma)$.

[4]$\text{ValueGap}(\pi) \leq n\epsilon_{\text{BC}}/(1-\gamma)^2$ from the Performance Difference Lemma (see e.g. Xiao (2022)) and $\text{ValueGap}(\pi) \leq \epsilon_\rho/(1-\gamma)$ by Hölder's inequality.

*When does exact occupancy measure matching learn a Nash equilibrium?*

We start by showing that under the strong assumption of full-state support, exact state-action occupancy measure matching (shortened state-action matching below) recovers an exact Nash equilibrium. Then, we show how relaxing any of these two assumptions can lead to catastrophic errors. Note that assuming exact state-action matching (i.e. $\epsilon_\rho = 0$) is equivalent to assuming state-only matching (i.e. $\epsilon_\mu = 0$) and exact Behavioral Cloning (i.e. $\epsilon_{\text{BC}} = 0$).

To make our statement more precise, note that any policy $\pi$ of a Markov Game partitions the state space $\mathcal{S}$ into a visited region $\mathcal{S}_\pi^+ = \{s : \mu_\pi(s) > 0\}$ and an unvisited region $\mathcal{S}_\pi^- = \{s : \mu_\pi(s) = 0\}$. We prove that state-action matching ($\epsilon_\rho = 0$) and full-state support ($\mathcal{S}_{\pi^E}^+ = \mathcal{S}$) recovers the Nash expert, i.e. $\pi = \pi^E$. When the state-support is incomplete ($\mathcal{S}_{\pi^E}^+ \neq \mathcal{S}$) or only state-matching ($\epsilon_\mu = 0$) holds, we can only guarantee the trivial bound $\text{NashGap}(\pi) \leq \mathcal{O}(1/(1-\gamma))$.

## 4.1 SUFFICIENCY OF STATE-ACTION MATCHING UNDER FULL-STATE SUPPORT

Under full-state support ($\mathcal{S}_\pi^+ = \mathcal{S}$), we show how state-action matching is sufficient to learn a Nash equilibrium. This is the direct consequence of the following fact: a policy $\pi$ is uniquely characterized by its state-action occupancy measure $\rho_\pi$ on the visited region $\mathcal{S}_\pi^+$. We formalize this idea in the following theorem, and explain its implications for measure matching.

**Theorem 1.** *Let $\pi, \pi' \in \Pi$ be such that $\rho_\pi = \rho_{\pi'}$. Then, $\mathcal{S}_\pi^+ = \mathcal{S}_{\pi'}^+$ and $\pi(\cdot|s) = \pi'(\cdot|s)$ for every $s \in \mathcal{S}_\pi^+$.*

The proof can be found in Appendix B.1.

This shows that in the limit of infinite data where state-action matching is attainable, the empirical state-action occupancy measure is a sufficient statistic for learning $\pi^E$ on its state support $\mathcal{S}_{\pi^E}^+$. Assuming full-state support, we can then derive the following corollary for state-action matching.

**Corollary 1.** *Let $\pi^E, \pi \in \Pi$ be such that $\mathcal{S}_{\pi^E}^+ = \mathcal{S}$ and $\rho_{\pi^E} = \rho_\pi$; then, $\text{NashGap}(\pi) = 0$.*

Matching state-action occupancy measure under full-state support is therefore a sufficient condition for learning the expert Nash equilibrium. In the next section, we show that only assuming state-only matching $\mu_{\pi^E} = \mu_\pi$ becomes insufficient to learn a Nash equilibrium.

## 4.2 INSUFFICIENCY OF STATE-ONLY MATCHING WITH FULL-STATE SUPPORT

In this section, we show that even with full-state support, state-only matching doesn't provide exploitability guarantees in general Markov Games. This is because rewards are functions of state-action pairs but state distributions are not necessarilly tied to specific transitions. We can therefore construct examples of games where a given state distribution can be realized by distinct transitions and very different rewards. To prove that state-only matching with full-state support cannot extract Nash equilibria (Nash gap of zero) in general games, we demonstrate below that it can even incur a Nash gap linear in the effective horizon $1/(1-\gamma)$.

**Lemma 1.** *There exists a game and a corresponding expert policy $\pi^E$ such that $\mathcal{S}_{\pi^E}^+ = \mathcal{S}$. Moreover, there exists a policy $\pi$ such that $\mu_{\pi^E} = \mu_\pi$ and $\text{NashGap}(\pi) \geq \Omega(1/(1-\gamma))$.*

*Proof.* We prove this lemma by constructing an example of such a game. Let $G$ be a cooperative two-player game with action sets $\mathcal{A}_1 = \mathcal{A}_2 = \{a_1, a_2\}$, state space $\mathcal{S} = \{s_0, s_1, s_2\}$, discount term $\gamma$, and uniform initial state distribution $\nu_U$. The rewards and transition dynamics of $G$ are shown in Figure 1. By definition, $\mu_{\pi'}(s) \geq (1-\gamma)\nu_U(s) > 0$ for all $s \in \mathcal{S}$ and $\pi' \in \Pi$. Therefore, all policies have full-state support.

A Nash equilibrium of $G$ is the constant policy $\pi^E((a_1, a_1)|s) = 1$ for all $s \in \mathcal{S}$ with uniform occupancy measure $\mu_{\pi^E}(\cdot) = 1/3$. This is a Nash equilibrium because mixed actions in this game always incur the worst possible value. A formal argument can be found in Appendix B.2.

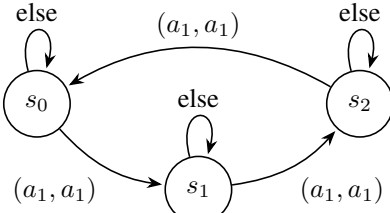
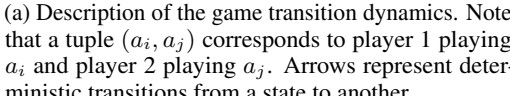
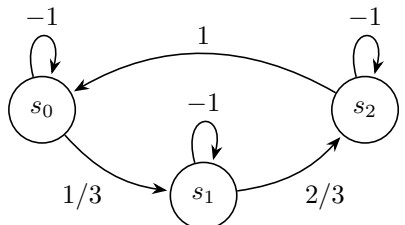

(a) Description of the game transition dynamics. Note that a tuple $(a_i, a_j)$ corresponds to player 1 playing $a_i$ and player 2 playing $a_j$. Arrows represent deterministic transitions from a state to another.

(b) Description of the reward function. The number on each arrow is the reward associated with the corresponding transition. The rewards are the same for both players.

Figure 1: Cooperative two-player game $G$: the transitions and rewards are described by the left and right sub-figures, respectively. In this game, there exists a Nash equilibrium $\pi^E$ with full-state support and a policy $\pi$ such that $\mu_{\pi^E} = \mu_\pi$ with a Nash gap linear in the effective horizon $1/(1-\gamma)$.

Let the learned policy be the constant $\pi((a_1, a_2)|s) = 1$ such that $\mu_\pi = \mu_{\pi^E}$ and $V_2^\pi(\nu_U) = -\frac{1}{1-\gamma}$. Noting that $V_2^{\pi^E}(\nu_U) = \frac{2/3}{1-\gamma}$, this concludes the proof as:

$$\text{NashGap}(\pi) \geq V_2^{\pi^E}(\nu_U) - V_2^\pi(\nu_U) = \frac{5/3}{1-\gamma} \geq \Omega\left(\frac{1}{1-\gamma}\right).$$

$\square$

Lemma 1 emphasizes that state-based occupancy approaches as in Wu et al. (2025) cannot guarantee convergence to equilibria in general games even when policies are guaranteed to have full state coverage.

### 4.3 INSUFFICIENCY OF STATE-ACTION MEASURE MATCHING WITH UNVISITED STATES

Assuming again state-action matching, we show that full-state support is essential for learning a Nash equilibrium in general games. We draw on the example given by Tang et al. (2024, Figure 2) to point out issues from a non-visited region $\mathcal{S}_{\pi^E}^- \neq \emptyset$ and derive the following theorem.

**Theorem 2.** *(Adapted from Tang et al. (2024, Theorem 4.3)) There exists a Markov Game with expert policy $\pi^E$ and a learned policy $\pi$ such that even if $\rho_{\pi^E} = \rho_\pi$, the Nash gap scales linearly with the discounted horizon; i.e., $\text{NashGap}(\pi) \geq \Omega(1/(1-\gamma))$.*

The key idea and intuition behind this theorem is that the imitation dataset misses information about the unvisited region $S_{\pi^E}^-$. An illustrative example of when this is undesirable is shown in Figure 2.

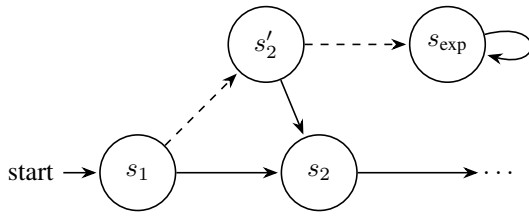

Figure 2: Transitions of a two-player Markov Game. The unique initial state is $s_1$. The rest of the chain ($\cdots$) and reward functions can be designed to induce linear Nash gap for state-action matching.

We can design reward functions for the transitions of Figure 2 such that the expert would always take the solid arrows.

The dataset missing information about $\pi^E(\cdot|s_2')$, a best-response can lead to states $s_2'$ then $s_{\exp}$. This last state $s_{\exp}$ can be designed to incur high rewards for one player, while the expert region $S_{\pi^E}^+$ incurs linear less.

For completeness, we adapt the proof of Tang et al. (2024) for infinite horizon games in Appendix B.3.

# 5 ON THE INFEASIBILITY OF TRACTABLE LOWER BOUNDS FOR EXPLOITABILITY

The analysis of Section 4 reveals that even under the idealized assumption of exact occupancy measure matching, MA-IL can still fail drastically. This prevents us from deriving *consistent* exploitability bounds in the general case. For practical settings, focusing on the idealized case of no learning error is however not sufficient as most IL procedures will incur an approximation error. This is due to two main reasons: First, because of the finite number of samples in the dataset, matching empirical statistics is inevitably different from matching the expert ones. Second, tbe function approximations in the non-tabular setting (Song et al., 2018; Wu et al., 2025) can only approximate the desired distributions.

This shift from an exact to an approximate matching regime introduces new challenges. We know that the value gap always enjoys a *consistent* and *tractable* upper bound of rate $\mathcal{O}\left(\epsilon_{\text{BC}}/(1-\gamma)^2\right)$ (Ross & Bagnell, 2010) or $\mathcal{O}(\epsilon_\rho/(1-\gamma))$. However, we will demonstrate in the current and the next sections that even small approximation errors can make exploitability bounds *intractable*.

A natural step in assessing the exploitability of the imitated policies is to quantify the best-case Nash gap they might induce under approximation errors. Given an approximation error, this quantity corresponds to the smallest achievable Nash gap among all the possible imitation policies $\pi$. Formally, we define it as follows:

**Definition 5** (Tight Nash gap lower bound). *Let $G$ be a Markov Game and let $\Pi^E$ the set of Nash equilibria of $G$. We define the tight Nash gap lower bound for $G$ with occupancy matching error $\epsilon_\rho$ as*

$$m_\rho(G, \epsilon_\rho) = \min_{\pi^E \in \Pi^E} \min_{\pi \in \mathcal{M}_{\epsilon_\rho}(\pi^E)} \text{NashGap}(\pi),$$

*where $\mathcal{M}_{\epsilon_\rho}(\pi^E) = \{\pi \in \Pi : \|\rho_\pi - \rho_{\pi^E}\|_1 = \epsilon_\rho\}$.*

For a game $G$, given only the approximation error assumption $\epsilon_\rho$, $m_\rho(G, \epsilon_\rho)$ corresponds to the best possible achievable Nash gap.

This quantity is in general intractable to compute for the case of bimatrix games as we show below in Theorem 3.

**Theorem 3.** *The problem of computing $m_\rho(G_{bi}, \epsilon_\rho)$ given an arbitrary bimatrix game $G_{bi}$ and an arbitrary $\epsilon_\rho \in \mathbb{R}_+$ is PPAD-hard[5].*

*Proof outline.* If computing this lower bound can be done efficiently, then it is also efficient to compute the support of a Nash equilibrium in a general bimatrix game. As we prove, the latter is PPAD-hard, so is our problem.

In the proof, we provide a polynomial reduction to the problem of computing a support of an approximate Nash equilibrium. Then, we show that this can be polynomially reduced to the problem of computing a Nash equilibrium itself.

This concludes the proof as finding an $\epsilon$-Nash equilibrium is a PPAD-complete problem (Chen et al., 2009). See Appendix B.4 for the formal proof. □

This theorem tells us that *evaluating* the best-case Nash gap for a given $\epsilon_\rho$ is not a tractable problem even when the game is fully known. This observation on the specific case of bimatrix games naturally extends to (infinite) repeated games and allows us to derive the following corollary on a larger class of Markov games.

**Corollary 2.** *The problem of computing $m_\rho(G, \epsilon_\rho)$ given an arbitrary Markov game $G$ and an arbitrary $\epsilon_\rho \in \mathbb{R}_+$ is PPAD-hard.*

We further note that the hardness results do not imply the impossibility of deriving analytical bounds, for example involving min-max optimization problems (that are known to be hard to compute (Daskalakis et al., 2021)). However, they imply that finding bounds with polynomial computation time is no easier than finding the Nash equilibria themselves, even if the game is known. This makes any potential tight lower bound on the Nash gap *intractable*.

---

[5]See Appendix A for more formal details on the PPAD complexity class.

# 6 TRACTABLE AND CONSISTENT EXPLOITABILITY UPPER BOUNDS FROM BEST-RESPONSE CONTINUITY

In the previous section, we worked on a lower bound to understand the best possible Nash gap that we can hope to achieve from given approximation errors. In this section, we study and characterize worst-case exploitability guarantees with a new notion of best-response continuity.

For general $n$-player Markov Games, the worst-case Value gap for a given BC error (Equation 2) is given by a *consistent* and *tractable* uniform bound: $\text{ValueGap}(\pi) \leq n\epsilon_{\text{BC}}/(1-\gamma)^2$. However, the exploitation nature of the Nash gap makes it impossible to derive such a bound for a fixed error term. We reveal how this phenomenon, not present in SA-IL, is characterized by a form of best-response continuity, leading to game-dependent bounds.

## 6.1 CHARACTERIZATION OF MARKOV GAMES VIA BEST-RESPONSE DELTA-CONTINUITY

We introduce below a notion of continuity of the best-response mapping of Markov Games that will allow us to derive new general upper bounds on the Nash gap in the next subsections.

**Definition 6** ($\delta$-continuity of the best-response correspondence)**.** *For a given game G, the best-response mapping is said to be $\delta$-continuous at equilibrium $\pi^E$ for some $\delta : \mathbb{R}^+ \to [0, 2]$ if for all $i \in [n]$, and $\epsilon > 0$, we have:*

$$\mathbb{E}_{s\sim\mu_{\pi^E}}\left[\left\|\pi_{-i}(\cdot|s) - \pi_{-i}^E(\cdot|s)\right\|_1\right] \leq \epsilon \implies \max_{\pi_i^*\in\text{BR}_i(\pi_{-i})} \mathbb{E}_{s\sim\mu_{\pi^E}}\left[\|\pi_i^*(\cdot|s) - \pi_i^E(\cdot|s)\|_1\right] \leq \delta(\epsilon).$$

This is a notion related to the maximal change over all $i \in [n]$ of the optimal policy $\pi_i^E \in \text{BR}_i(\pi_{-i}^E)$, when the induced MDP for player $i$ is produced by $\pi_{-i}$ instead of the equilibrium behavior $\pi_{-i}^E$. This continuity will be used below to reduce the complexity of computing Nash gap upper bounds to computing a valid $\delta$. This definition is naturally extended for a class of games as follows.

**Definition 7** ($\delta$-continuity of a class of games)**.** *A class $\mathcal{C}$ of Markov Games is $\delta$-continuous if every game $G \in \mathcal{C}$ is $\delta$-continuous at all its Nash equilibria.*

Provided with a $\delta$-continuous class of games, we will therefore be able to derive bounds without assuming a specific game. However, we note that even for the class of games with a *consistent* bound as presented in Section 4, we cannot guarantee more than the trivial $\delta(\epsilon) = 2$ for $\epsilon > 0$.

**Lemma 2.** *Let $\mathcal{C}$ be the class of games with consistent bounds. This class is $\delta$-continuous only for trivial $\delta$ such that $\delta(\epsilon) = 2$ for all $\epsilon > 0$.*

*Proof.* Suppose $\mathcal{C}$ is $\delta$-continuous. For every $\epsilon > 0, \gamma \in (0, 1)$, we show that there exists a two-player game $G \in \mathcal{C}$ with Nash equilibrium $\pi^E$ where $\epsilon_{\text{BC}} \leq \epsilon$ can incur $\mathbb{E}_{s\sim\mu_{\pi^E}}\left[\left\|\pi_1^*(\cdot|s) - \pi_1^E(\cdot|s)\right\|_1\right] = 2$ for some $\pi_1^* \in \text{BR}_1(\pi_2)$.

We let $M_k$ be a chain of $k = \left\lceil \frac{\log(\frac{\epsilon}{2}(1-\gamma))}{\log(\gamma)} \right\rceil$ consecutive states and define its transitions as follows.

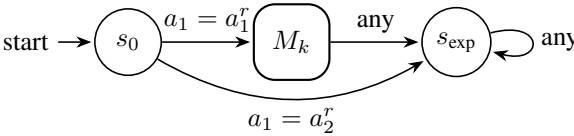

Figure 3: Deterministic transition dynamics of a two-player game $G$, with states $s_0$, $s_{\text{exp}}$ and sub-Markov chain $M_k$. Player 1 has action space $\mathcal{A}_1 = \{a_1^r, a_2^r\}$ and player 2 has action space $\mathcal{A}_2 = \{a_1^c, a_2^c\}$.

The rewards $r_1, r_2$ are independent of $a_1$ and defined as:

$$r_1(\cdot, a_1^c, s_{\text{exp}}) = 1, \qquad r_2(\cdot, a_2^c, s_{\text{exp}}) = 1, \qquad r_1(\cdot, a_2^c, s_{\text{exp}}) = -1,$$

and $r_1 = r_2 = 0$ otherwise.

A Nash equilibrium of this game is the constant policy $\pi^E((a_1^r, a_2^c)|s) = 1$ for all $s \in \mathcal{S}$. The expert is such that $\mu_{\pi^E}(s_{\exp}) \leq \epsilon/2$. Therefore, the policy $\pi((a_1, a_2)|s) = \pi^E((a_1, a_2)|s)$ if $s \neq s_{\exp}$ and $\pi((a_1^r, a_1^c)|s_{\exp}) = 1$, has BC error at most $\epsilon$.

A best-response to $\pi_2$ is the constant policy $\pi_1^*(a_2^r|s) = 1$ for all $s \in \mathcal{S}$ which incurs $\mathbb{E}_{s \sim \mu_{\pi^E}} \left[ \left\| \pi_1^*(\cdot|s) - \pi_1^E(\cdot|s) \right\|_1 \right] = 2$. Note that $G$ has a *consistent* bound for any error assumption ($\epsilon_{BC} = 0 \Leftrightarrow \epsilon_\mu = 0 \Leftrightarrow \epsilon_\rho = 0$ for $G$, and $\mathcal{S}_{\pi^E}^+ = \mathcal{S}$). $\qquad\square$

## 6.2 PROVABLE CONVERGENCE UNDER STRATEGIC DOMINANCE

The previous section showed that even for the class of games with consistent bounds, we cannot do better than the trivial $\delta$-continuity where $\delta(\epsilon) = 2$ for $\epsilon > 0$. We study in this section the other extreme for $\delta$: the constant $\delta(\cdot) = 0$, before studying the general case in the next section.

In fact, we show that this special case corresponds to the class of Dominant Strategy Equilibria, for which we can thus derive *consistent* and *tractable* exploitability upper bounds.

**Definition 8** (Dominant Strategy Equilibrium (DSE)). *A policy $\pi^E$ is a (weak) dominant strategy equilibrium if for every player $i$, $\pi_i^E$ is a weak dominant strategy, i.e.:*

$$V_i^{\pi_i^E, \pi_{-i}}(\nu_0) \geq V_i^{\pi_i, \pi_{-i}}(\nu_0) \quad \forall \pi \in \Pi.$$

Note that the key property induced by the DSE assumption is that $\pi_i^E$ is a best-response policy to any deviations $\pi_{-i}$ for every player $i$, which corresponds to $\delta$-continuity of the best-response for $\delta(\cdot) = 0$. This allows us to derive the following *consistent* and *tractable* upper bound on the Nash gap.

**Lemma 3.** *Suppose $\pi^E$ is a (weak) Dominant Strategy Equilibrium. Then, any learned policy $\pi$ with BC error $\epsilon_{BC}$ satisfies* $\mathrm{NashGap}(\pi) \leq 2n\epsilon_{BC}/(1-\gamma)^2$.

*Proof outline.* We leverage the fact that the Dominant Strategy Equilibrium assumption removes the ambiguity in how far the best-response of any player is from its individual expert policy, i.e. we have:

$$\mathrm{NashGap}(\pi) = \max_i \left[ V_i^{\pi_i^E, \pi_{-i}}(\nu_0) - V_i^{\pi_i, \pi_{-i}}(\nu_0) \right]. \tag{3}$$

Using Equation 3, we can add and subtract the expert value $V_i^{\pi^E}(\nu_0)$ for every $i \in [n]$ and apply the performance difference lemma (see e.g. Xiao (2022, Lemma 1)) twice to get:

$$\mathrm{NashGap}(\pi) \leq \frac{1}{(1-\gamma)^2} \cdot \max_i \mathbb{E}_{s \sim \mu_{\pi^E}} \left[ \left\| \pi_{-i}(\cdot|s) - \pi_{-i}^E(\cdot|s) \right\|_1 + \left\| \pi(\cdot|s) - \pi^E(\cdot|s) \right\|_1 \right].$$

We conclude the proof using the definition of the BC error and the fact that both $\pi$ and $\pi^E$ are product policies. The formal proof is deferred to Appendix C.1 $\qquad\square$

Note that when $n = 2$, this recovers the upper bound of Freihaut et al. (2025) with a fixed BC error.

Providing such an upper bound allows assessing an imitation policy a posteriori, given a specific BC error. As a bound polynomial in the game parameters, we consider it to be *tractable*. Alternatively, we can interpret the bound as a criterion on the BC error to ensure a fixed Nash approximation error.

**Corollary 3.** *Suppose $\pi^E$ is a (weak) Dominant Strategy Equilibrium; then, the recovered behavioral cloning policy $\pi$ is an $\epsilon$-Nash equilibrium if $\epsilon_{BC} \leq \frac{\epsilon(1-\gamma)^2}{2n}$.*

*Proof.* This is a direct consequence of Lemma 3 derived by inverting the Nash gap bound. $\qquad\square$

## 6.3 RELAXING THE DOMINANCE ASSUMPTION

Using a similar proof technique, we extend Lemma 3 by assuming general best-response $\delta$-continuity. This is demonstrated in the following lemma which we prove in Appendix C.2.

**Lemma 4.** *Suppose the equilibrium expert is $\pi^E$ and the game is $\delta$-continuous at $\pi^E$. Then,* $\mathrm{NashGap}(\pi) \leq \frac{2n\epsilon_{BC} + \delta(\epsilon_{BC})}{(1-\gamma)^2}$.

This is a generalization of Lemma 3 where the DSE case is recovered by setting $\delta(\cdot) = 0$. A numerical validation of the bound is provided in Appendix E.

Note that we do not claim that the above bound is tight due to our worse-case guarantee. However, this characterization of the Nash gap with $\delta$-continuity offers several advantages.

**Properties of the bound.** Consistence and tractability properties of the bound are directly related to the properties of the $\delta$ function: the bound is *consistent* if $\delta(0) = 0$ and it is *tractable* if $\delta$ itself is tractable. Lemma 4 then provides the key insight that deriving exploitability upper bounds reduces to characterizing $\delta$ for the considered game.

**On the sensitivity of game equilibria.** In comparison to the previous bound of Freihaut et al. (2025) obtained via a change of measure, our bound doesn't necessarily become vacuous if strategic deviations on the imitation policy explores regions of the state space not visited by the expert. In fact, $\delta$-continuity of games serves as a spectrum to characterize their sensitivity to perturbations at equilibrium points.

The first extreme case is $\delta(\cdot) = 0$ that corresponds to games with only dominant strategy Nash equilibria. This assumption essentially reduces the Nash gap computation to a Value gap, recovering the known bound $\text{ValueGap}(\pi) \leq n\epsilon_{\text{BC}}/(1 - \gamma)^2$ for single-agent behavioral cloning[6] (Ross et al., 2011) up to a constant factor 2. The other extreme is $\delta$-continuity with the constant $\delta(\cdot) = 2$, reducing to a trivial upper bound greater than $2/(1 - \gamma)$ and achieved by the limit of pathological games, as shown with the example given in the proof of Lemma 2.

Intuitively, a well-behaving $\delta$ can be imposed by regularizing the game, essentially smoothing the best-response map by promoting exploration (Ahmed et al., 2019; Geist et al., 2019). Furthermore, we note that large discontinuities in $\delta$ are favored by high variance in the expert rewards, where small deviations from the expert can induce large differences in expected rewards. Similarly, these high variations at the equilibrium can be penalized by risk-aversion (Mazumdar et al., 2024), and we hypothesize that it could lead to the derivation of better $\delta$ functions. We provide additional numerical validations of the impact of entropy regularization on $\delta$-continuity in Appendix E.

## 7 CONCLUSION

In this work, we consider the problem of learning a Nash equilibrium from a given dataset of expert demonstrations in a multi-agent system. Assuming a Nash equilibrium expert and a given imitation learning error (BC or measure matching), we study the derivation of both *consistent* and *tractable* guarantees on the Nash gap of the learned policy. In the idealized case of exact measure matching, we demonstrate that only full-state support and state-action matching can guarantee non-trivial Nash gaps. Moving to practical settings, we show how approximation errors introduce challenges that are not present in the single-agent case. For behavioral cloning, we then introduce the notion of delta-continuity related to strategy dominance, and show how this can be used to bound exploitability of the learned policy.

Looking forward, we see reachability assumptions and policy distribution-norms (Wei et al., 2017; Maillard et al., 2014) as good candidates for tighter game-dependent bounds. A potential improvement might also be achieved from the data part, by augmenting expert demonstrations with suboptimal trajectories (e.g. in SA-IL (Kim et al., 2021)), inspired by online IL (Ross et al., 2011; Freihaut et al., 2025) and unilateral deviation assumptions (Cui & Du, 2022).

ACKNOWLEDGMENTS

This work was supported in part by the National Science Foundation (NSF) under grants number ECCS-2438314 (CAREER Award) and CCF-2423134, the Army Research Laboratory (ARL) under grant number W911NF-26-1-0002, and institutional support from EPFL.

---

[6]The joint policy $\pi$ can be viewed as a single-agent policy with behavioral cloning loss (Equation 2) upper bounded by $n\epsilon_{\text{BC}}$ (see Lemma D.2 in Appendix).

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

# A ADDITIONAL BACKGROUND

## A.1 PPAD COMPLEXITY CLASS

In computational complexity theory, problems are categorized into classes to formally reason about their inherent difficulty and whether they are likely to be tractable. In game theory, the NP and PPAD classes are of crucial importance, as it has been shown that computing Nash equilibria is PPAD-complete (Daskalakis et al., 2009; Chen et al., 2009) and many decision problems around Nash equilibria are NP-hard (Conitzer & Sandholm, 2002). These key results also reinforce the arguments under which finding expert policies without demonstrations can be computationally intractable.

We define below the PPAD class introduced by Papadimitriou (1994) and related to some of our negative results.

**Definition 9** (PPAD class). *A search problem $\Pi$ belongs to the complexity class PPAD (Polynomial Parity Arguments on Directed graphs) if and only if it is polynomial-time reducible to the End-of-the-Line problem defined as follows:*

> ### End-of-the-Line Problem
>
> INPUT:
>
> - *A directed graph $G = (V, E)$ implicitly represented by two polynomial-time computation mutually inverse functions $P$ and $S$:*
>   - *$P : V \to V$ maps every vertex $v \in V$ to its unique predecessor, or itself.*
>   - *$S : V \to V$ maps every vertex $v \in V$ to its unique successor, or itself.*
> - *A source vertex $s \in V$ such that $P(s) = s$ and $S(s) \neq s$.*
>
> OUTPUT: *Either a sink vertex or another source vertex.*

PPAD problems belong to the larger TFNP class (Total Function NP), containing search problems for which a solution is guaranteed to exist (the problems are said to be total). The fundamental property that makes PPAD problems total (guaranteed to have a solution) is based on the parity argument: in any directed graph where each vertex has at most one incoming and one outgoing edge, if there exists a source, there must exist either another source or a sink. Therefore, the End-of-the-Line problem always has a solution.

It is believed that PPAD is not part of P, and hence PPAD-hard problems are believed intractable.

## A.2 GAIL AND OCCUPANCY-MEASURE MATCHING

Generative Adversarial Imitation Learning introduced by Ho & Ermon (2016) equivalently solves the following Inverse Reinforcement Learning (IRL) problem

$$\mathrm{IRL}_\psi(\pi^E) = \arg \max_{c \in \mathcal{C}} -\psi(c) + \left( \min_{\pi \in \Pi} -H(\pi) + \mathbb{E}_\pi[c(s, a)] \right) - \mathbb{E}_{\pi^E}[c(s, a)]$$

for a cost $c \in \mathcal{C}$ followed by standard Reinforcement Learning (RL) for policy extraction

$$\mathrm{RL}(c) = \arg \min_{\pi \in \Pi} -H(\pi) + \mathbb{E}_\pi[c(s, a)],$$

with sets $\mathcal{C}, \Pi$ constrained by modelization expressivity, $\psi : \mathbb{R}^{\mathcal{S} \times \mathcal{A}} \to \mathbb{R} \cup \{\infty\}$ a convex cost function regularizer, and $H(\pi) = \mathbb{E}_\pi[-\log \pi(a|s)]$ the causal entropy of policy $\pi$. Their key innovation is to show that for a particular instance of $\psi$, both problems can be solved simultaneously by training a discriminative classifier $D : \mathcal{S} \times \mathcal{A} \to (0, 1)$ and a generator policy $\pi \in \Pi$ in a GAN-like (Goodfellow et al., 2020) manner. For a non-restricted $\mathcal{C} = \mathbb{R}^{\mathcal{S} \times \mathcal{A}}$, we can exchange the max-min for a min-max (Ho & Ermon, 2016; Garg et al., 2021) and end up with the following practical optimization formulation:

$$\min_{\pi \in \Pi} \max_{D \in (0,1)^{\mathcal{S} \times \mathcal{A}}} \mathbb{E}_\pi[\log(D(s, a))] + \mathbb{E}_{\pi^E}[\log(1 - D(s, a))] - \lambda H(\pi),$$

where $\lambda$ is a regularization factor.

This problem in particular is shown to be equivalent to the following regularized state-action occupancy matching objective: $\min_\pi \mathrm{D_{JS}}(\rho_\pi, \rho_{\pi^E}) - \lambda H(\pi)$, with $\mathrm{D_{JS}}$ denoting the Jensen-Shannon divergence.

In the more general case, Ho & Ermon (2016) propose that imitation learning can be done by state-action occupancy measure matching problems of the form:

$$\min_\pi \psi^*(\rho_\pi - \rho_{\pi^E}) - H(\pi)$$

where the entropy regularization makes the optimal BC policy unique and $\psi^*$ denotes the convex conjugate of $\psi$.

## B  PROOFS OF HARDNESS RESULTS

### B.1  PROOF OF THEOREM 1

**Theorem 1.** *Let $\pi, \pi' \in \Pi$ be such that $\rho_\pi = \rho_{\pi'}$. Then, $\mathcal{S}_\pi^+ = \mathcal{S}_{\pi'}^+$ and $\pi(\cdot|s) = \pi'(\cdot|s)$ for every $s \in \mathcal{S}_\pi^+$.*

*Proof.* Let $\pi, \pi'$ be two policies such that $\rho_\pi = \rho_{\pi'}$.

We will first prove that $\mathcal{S}_\pi^+ = \mathcal{S}_{\pi'}^+$ then prove that the policies are equal on the visited region $\mathcal{S}_\pi^+$.

1) The policies $\pi, \pi'$ visit the same region of the space state.

Suppose towards contradiction that there exists some $s \in \mathcal{S}$ such that $s \in \mathcal{S}_\pi^+$ and $s \notin \mathcal{S}_{\pi'}^+$.

Since $\pi(\cdot|s)$ is a probability distribution, there must be an action $a \in \mathcal{A}$ such that $\pi(a|s) > 0$.

Hence $\rho_\pi(a, s) > 0$ but $\rho_{\pi'}(a, s) = \mu_{\pi'}(s)\pi'(a|s) = 0$ by assumption.

This is a contraction since $\rho_\pi = \rho_{\pi'}$, hence $\mathcal{S}_\pi^+ = \mathcal{S}_{\pi'}^+$.

2) The policies $\pi, \pi'$ are equal on their shared visited region.

We again proceed with a proof by contradiction.

Suppose there exists $a \in \mathcal{A}$ and $s \in \mathcal{S}_\pi^+$ such that $\pi(a|s) \neq \pi'(a|s)$. Since $\rho_\pi(a, s) = \rho_{\pi'}(a, s)$, $\mu_\pi(s) > 0, \mu_{\pi'}(s) > 0$ and $\pi(a|s) \neq \pi'(a|s)$, we must therefore have $\mu_\pi(s) \neq \mu_{\pi'}(s)$.

Without loss of generality, assume $\mu_\pi(s) > \mu_{\pi'}(s)$; thus, for all $a' \in \mathcal{A}$, $\pi(a'|s) < \pi'(a'|s)$ by the equality of the state-action occupancy measures.

This is a contradiction since that would mean

$$\sum_{a' \in \mathcal{A}} \pi(a'|s) < \sum_{a' \in \mathcal{A}} \pi'(a'|s) = 1$$

but $\pi(\cdot|s)$ is a probability distribution. $\square$

### B.2  PROOF OF NASH EQUILIBRIA FOR THE GAME IN FIGURE 1

We want to show that $\pi^E((a_1, a_1)|s) = 1$ for all $s \in \mathcal{S}$ is indeed a Nash equilibrium of the two-player game described in figure Figure 1.

Suppose towards contradiction that $\pi^E$ is not a Nash equilibrium, as some player $i \in \{1, 2\}$ is better off from unilaterally deviating. By the performance difference lemma (Lemma D.1), there must exists a state $s \in \mathcal{S}$ with positive advantage

$$A_i^{\pi^E}(a^i, s) = Q_i^{\pi^E}(s, a^i, a^{-i}) - V_i^{\pi^E}(s), \quad \text{with } a^{-i} = a_1$$

when the player $i$ chooses $a^i = a_2$ and $Q_i^{\pi^E}(s, a^i, a^{-i}) := r_i(s, a^i, a^{-i}) + \gamma \sum_{s' \in \mathcal{S}} P(s'|s, a) V_i^{\pi^E}(s')$.

This is a contradiction since, when fixing $a^i = a_2$, for all $s \in \mathcal{S}$ we have

$$
\begin{aligned}
A_i^{\pi^E}(a^i, s) &= r_i(a^i \neq a^{-i}, s) - V_i^{\pi^E}(s) + \gamma Q_i^{\pi^E}(s, a^i, a^{-i}) \\
&= -1 - (1-\gamma)V_i^{\pi^E}(s) + \gamma A_i^{\pi^E}(a^i, s) \\
&\leq \gamma A_i^{\pi^E}(a^i, s)
\end{aligned}
$$

Which is a contradiction as we assumed $A_i^{\pi^E}(a^i, s) > 0$. Hence $\pi^E$ is a Nash equilibrium.

### B.3   PROOF OF THEOREM 2

**Theorem 2.** *(Adapted from Tang et al. (2024, Theorem 4.3)) There exists a Markov Game with expert policy $\pi^E$ and a learned policy $\pi$ such that even if $\rho_{\pi^E} = \rho_\pi$, the Nash gap scales linearly with the discounted horizon; i.e., $\mathrm{NashGap}(\pi) \geq \Omega\left(1/(1-\gamma)\right)$.*

The proof below has been extracted from Tang et al. (2024) and only slightly adapted for infinite horizon games.

*Proof.* We explicitly construct a two-player common payoff Markov Game with infinite horizon and a common action space $\mathcal{A}_i = \{a_1, a_2, a_3\}$ for each agent $i = 1, 2$. The state space $\mathcal{S}$ is countably infinite and ordered.

The action-independent (shared) reward function is defined as $r(s_i) = 1$ if $i$ is odd, and $r(s_i) = 0$ if $i$ is even. The transition dynamics are described in Figure 4.

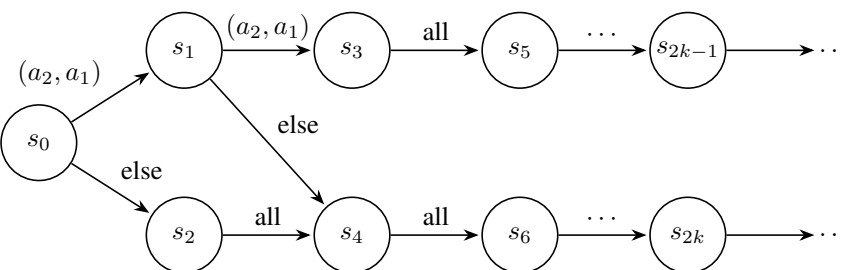

Figure 4: Transition dynamics for the two-player game. $s_0$ is the initial state. The top branch contains all odd states while the bottom branch together with $s_0$ comprises all even states.

We define the expert as the Nash policy $\pi^E$ such that $\pi^E((a_1, a_1)|s_0) = 1$, $\pi^E((a_3, a_3)|s_1) = 1$ and the actions for the other states are arbitrary.

Similarly, we define the trained policy $\pi$ such that $\pi((a_1, a_1)|s_0) = 1$ and $\pi((a_1, a_1)|s_1) = 1$, and plays the same as the expert on the other states.

In this case $\rho_\pi = \rho_{\pi^E}$ but

$$
\mathrm{NashGap}(\pi) \geq V_1^{\pi_1', \pi_2}(s_0) - V_1^\pi(s_0) = \frac{1}{1-\gamma} - 1 \geq \Omega\left(\frac{1}{1-\gamma}\right)
$$

where

$$
\pi_1'(a^1|s) = \begin{cases} 1 & \text{if } a^1 = a_2 \text{ and } s \in \{s_0, s_1\} \\ 0 & \text{if } a^1 \neq a_2 \text{ and } s \in \{s_0, s_1\} \\ \pi_1(a^1|s) & \text{otherwise} \end{cases}
$$

$\square$

## B.4 PROOF OF THEOREM 3

**Theorem 3.** *The problem of computing $m_\rho(G_{bi}, \epsilon_\rho)$ given an arbitrary bimatrix game $G_{bi}$ and an arbitrary $\epsilon_\rho \in \mathbb{R}_+$ is PPAD-hard[7].*

Before proving Theorem 3, let us prove the following intermediary lemma. This will allow us to conclude by doing a reduction to the problem of finding an $\epsilon$-Nash equilibrium.

**Lemma 5.** *Finding an $\epsilon$-Nash equilibrium support in a general bimatrix game is PPAD-complete.*

### B.4.1 PROOF OF LEMMA 5

Before proving Theorem 5, we introduce below a simpler version for exact Nash equilibria.

**Theorem 4.** *Finding the support of any Nash equilibrium in a bimatrix game is PPAD-complete.*

*Proof.* We prove this by a polynomial reduction from the problem of finding the equilibrium in a bimatrix game to the problem of finding its support. The other direction is trivial.

Let $\mathcal{G} = (A_1, A_2)$ be a bimatrix game with payoff matrices $A_i$ for each player $i \in \{1, 2\}$. For simplicity we further assume that $A_1, A_2 \in \mathbb{R}^{n \times n}$ (i.e. both players have the same number of actions $n$).

Let $\pi_1, \pi_2$ the unknown policies at equilibrium of our game, with respective supports $\nu_1, \nu_2 \subseteq \{1, \ldots, n\}$. Using our support finding oracle, we compute $\nu_1, \nu_2$ from the payoff matrices.

We note that from the definition of a Nash equilibrium, $\pi_1$ is a best-response of player 1 to player 2 having policy $\pi_2$. This means:

$$\pi_1^\top A_1 \pi_2 = \max_i A_{1,i}^\top \pi_2 \implies A_{1,j}^\top \pi_2 = A_{1,k}^\top \pi_2 \; \forall j, k \in \nu_1,$$

and a similar argument holds for $\pi_2$, this is known as the indifference principle.

Using this, we can rewrite the problem of finding a Nash equilibrium as follows:

$$
\begin{aligned}
\text{Find} \quad & x \in \mathbb{R}^{|\nu_1|}, \; y \in \mathbb{R}^{|\nu_2|} \\
\text{s.t.} \quad & \sum_{i=1}^{|\nu_1|} x_i = \sum_{i=1}^{|\nu_2|} y_i = 1 \\
& (A_{2,j} - A_{2,k})^\top x = 0 \quad \forall j, k \in \nu_2 \\
& (A_{1,j} - A_{1,k})^\top y = 0 \quad \forall j, k \in \nu_1 \\
& x_i \geq 0 \quad \forall i \in \nu_1 \\
& y_i \geq 0 \quad \forall i \in \nu_2
\end{aligned}
$$

This is a linear program that can be solved in polynomial time. By solving this optimization problem we recover $\pi_1 = x$ and $\pi_2 = y$ (see also Algorithm 3.4 in von Stengel), which is valid with our assumptions of equilibrium supports $\nu_1, \nu_2$.

However, as shown in Chen et al. (2009), finding an equilibrium of a bimatrix game is PPAD-complete. This concludes our proof by showing that finding the support of a Nash equilibrium in general bimatrix games is PPAD-complete. $\square$

Using similar arguments, we argue that the theorem also holds for epsilon Nash equilibria with the following proof.

*Proof of Lemma 5.* The proof follows by adapting the linear program used in the proof of Theorem 4, replacing the equality constraints due to the indifference principle by two inequality constraints allowing some slackness of magnitude less $\epsilon$ as follows

---

[7]See Appendix A for more formal details on the PPAD complexity class.

$$\text{Find} \quad x \in \mathbb{R}^{|\nu_1|}, \ y \in \mathbb{R}^{|\nu_2|}$$

$$\text{s.t.} \quad \sum_{i=1}^{|\nu_1|} x_i = \sum_{i=1}^{|\nu_2|} y_i = 1$$

$$(A_{2,j} - A_{2,k})^\top x \le \epsilon \quad \forall j, k \in \nu_2$$

$$(A_{1,j} - A_{1,k})^\top y \le \epsilon \quad \forall j, k \in \nu_1$$

$$(A_{2,k} - A_{2,j})^\top x \le \epsilon \quad \forall j, k \in \nu_2$$

$$(A_{1,k} - A_{1,j})^\top y \le \epsilon \quad \forall j, k \in \nu_1$$

$$x_i \ge 0 \quad \forall i \in \nu_1$$

$$y_i \ge 0 \quad \forall i \in \nu_2$$

This allows finding an $\epsilon$-Nash equilibrium which is also known to be PPAD-complete (Chen et al., 2009). The other direction is trivial. □

### B.4.2 REDUCTION FOR THE PROOF OF THEOREM 3

Recall the statement to prove

**Theorem 3.** *The problem of computing $m_\rho(G_{bi}, \epsilon_\rho)$ given an arbitrary bimatrix game $G_{bi}$ and an arbitrary $\epsilon_\rho \in \mathbb{R}_+$ is PPAD-hard[8].*

*Proof.* We prove the result by a polynomial reduction from the problem of finding the support of an $\epsilon$-Nash equilibrium in a general bimatrix game to the problem of computing $m_\rho(G_{bi}, \epsilon_\rho)$ for any $G_{bi}, \epsilon_\rho$. Note that in this one-state game, the state-action occupancy measure is equal to the policy distribution. We assume $\max_{i,j} \max\{|A_{1i,j}|, |A_{2i,j}|\} < 1$. If it's not the case, it suffices to divide the payoff matrices by 2 and apply the same argument for $\epsilon_\rho/2$.

Assume access to an oracle $L(G_{bi}, \epsilon)$ for $m_\rho$ for any $G_{bi} = (A_1, A_2), \epsilon_\rho \in \mathbb{R}_+$. We will show that polynomially many calls to this oracle are sufficient to find the support of any $\epsilon$-Nash equilibrium of $G_{bi}$. Specifically, Algorithm 1 described below is sufficient for this task.

To prove this fact, it suffices to note that replacing $A_k^\nu$ by $A_k^i$ for any $k \in \{1, 2\}, i \in [n]$ changes the value of the lower bound only if $\pi_{k,i} > 0$ for every $s$-Nash equilibrium $\pi$ with supports supersets of $\nu_1, \nu_2$.

Formally, let $\Pi_{\min}^s$ be the set of policies minimizing the Nash gap with the imitation error $\epsilon$:

$$\Pi_{\min}^s := \arg\min_{\pi_1, \pi_2 : \|\pi_1 \pi_2^\top - \pi_1^E \pi_2^{E\top}\|_1 = s} \text{NashGap}(\pi_1, \pi_2),$$

and arrange the set in lexicographic order of the concatenated indicator vector encodings of the supports of the two players. We show that $(\nu_1, \nu_2)$ is equal to the first element of $\Pi_{\min}^s$ which we denote $(\nu_1^0, \nu_2^0)$.

Let $j \in \{1, 2\}$ be a fixed player, then for every $i \in [n]$:

- Case $i \in \nu_j$: then there doesn't exist any element with a smaller lexicographic index in $\Pi_{\min}^s$. Hence, $\nu_j \subseteq \nu_j^0$.

- Case $i \notin \nu_j$: then there must be an equilibrium such that player $j$ takes action $i$ with null probability. Hence, $\nu_j^0 \subseteq \nu_j$

Thus, $\nu_j = \nu_j^0$ and $(\nu_1, \nu_2)$ are valid supports for an $\epsilon$-Nash equilibrium.

Applying Lemma 5 allows us to conclude that finding $L$ is a PPAD-hard problem. □

---

[8]See Appendix A for more formal details on the PPAD complexity class.

---

**Algorithm 1** Lower bound reduction

---

Given a game $A_1, A_2$ and a target Nash precision $\epsilon$.
Define $A_1^\nu = A_1$, $A_2^\nu = A_2$.
Define $K$ such that $-1 < K < -\max_{i,j}\max\{|A_{1i,j}|, |A_{2i,j}|\}$.
Define $\nu_1 = \nu_2 = \emptyset$.
Define $\delta = \|\epsilon/K \cdot e_1\|_1$ such that $s = L((A_1, A_2), \delta) \le \epsilon$, for a canonical vector $e_1$.
**for** $i \in \{1, \ldots, n\}$ **do**
    Define $A_1^i$ such that

$$\bullet \ \{A_1^i\}_{a,b} = \{A_1^\nu\}_{a,b} \ \forall a, b \in [n] \setminus \{i\} \times [n]$$

$$\bullet \ \{A_1^i\}_i = K \cdot 1$$

    Use the oracle to compute $l_1^i = L((A_1^i, A_2), \delta)$.
    **if** $l_1^i > s$ **then**
       |   $\nu_1 \leftarrow \nu_1 \cup \{i\}$.
    **else**
        $A_1^\nu \leftarrow A_1^i$.

**for** $i \in \{1, \ldots, n\}$ **do**
    Define $A_2^i$ such that

$$\bullet \ \{A_2^i\}_{a,b} = \{A_2^\nu\}_{a,b} \ \forall a, b \in [n] \setminus \{i\} \times [n]$$

$$\bullet \ \{A_2^i\}_i = K \cdot 1$$

    Use the oracle to compute $l_2^i = L((A_1, A_2^i), \delta)$.
    **if** $l_2^i > s$ **then**
       |   $\nu_2 \leftarrow \nu_2 \cup \{i\}$.
    **else**
        $A_2^\nu \leftarrow A_2^i$.

Return $\nu_1, \nu_2$

---

## C  PROOFS OF UPPER BOUNDS

### C.1  PROOF OF LEMMA 3

**Lemma 3.** *Suppose $\pi^E$ is a (weak) Dominant Strategy Equilibrium. Then, any learned policy $\pi$ with BC error $\epsilon_{BC}$ satisfies* $\mathrm{NashGap}(\pi) \le 2n\epsilon_{BC}/(1-\gamma)^2$.

*Proof.* Following the proof outline from the main text, we use the Dominant Strategy Equilibrium assumption to simplify the Nash gap as follows:

$$\mathrm{NashGap}(\pi) = \max_{i,\pi_i^*}\left[V_i^{\pi_i^*,\pi_{-i}}(\nu_0) - V_i^{\pi_i,\pi_{-i}}(\nu_0)\right] = \max_i\left[V_i^{\pi_i^E,\pi_{-i}}(\nu_0) - V_i^{\pi_i,\pi_{-i}}(\nu_0)\right]$$

We can then add and subtract the expert value for every player, and apply the performance difference lemma (Lemma D.1) twice:

$$\mathrm{NashGap}(\pi) = \max_i\left[V_i^{\pi_i^E,\pi_{-i}}(\nu_0) - V_i^{\pi_i,\pi_{-i}}(\nu_0)\right]$$

$$= \max_i\left[V_i^{\pi_i^E,\pi_{-i}}(\nu_0) - V_i^{\pi^E}(\nu_0) + V_i^{\pi^E}(\nu_0) - V_i^{\pi_i,\pi_{-i}}(\nu_0)\right]$$

$$\le \frac{1}{1-\gamma}\cdot\max_i\mathbb{E}_{s\sim\mu_{\pi^E}}\left[\sum_a Q^\pi(s,a)\left(\pi^E(a|s) - \pi(a|s)\right)\right]$$

$$- \frac{1}{1-\gamma}\cdot\max_i\mathbb{E}_{s\sim\mu_{\pi^E}}\left[\sum_a Q^{\pi_i^E,\pi_{-i}}(s,a)\left(\pi_{-i}^E(a_{-i}|s) - \pi_{-i}(a_{-i}|s)\right)\pi_i^E(a_i|s)\right]$$

where the Q-functions are defined as in Lemma D.1.

Now upper bounding the Q-functions using $\|Q^{\pi'}\|_\infty \le 1/(1-\gamma)$ for any policy $\pi' \in \Pi$:

$$\mathrm{NashGap}(\pi) \le \frac{1}{(1-\gamma)^2} \cdot \max_i \mathbb{E}_{s\sim\mu_{\pi^E}}\left[\left\|\pi(\cdot|s) - \pi^E(\cdot|s)\right\|_1 + \left\|\pi_{-i}(\cdot|s) - \pi^E_{-i}(\cdot|s)\right\|_1\right]$$

Using Lemma D.2 we conclude,

$$\mathrm{NashGap}(\pi) \le \frac{(2n-1)\epsilon_{\mathrm{BC}}}{(1-\gamma)^2} \le \frac{2n\epsilon_{\mathrm{BC}}}{(1-\gamma)^2}$$

$\square$

## C.2 Proof of Lemma 4

**Lemma 4.** *Suppose the equilibrium expert is $\pi^E$ and the game is $\delta$-continuous at $\pi^E$. Then,* $\mathrm{NashGap}(\pi) \le \frac{2n\epsilon_{BC}+\delta(\epsilon_{BC})}{(1-\gamma)^2}$.

The key is to use a very similar construction as for the proof of Lemma 3, leveraging the triangle inequality to introduce the slackness $\delta(\epsilon_{\mathrm{BC}})$.

*Proof.* Using similar arguments, we apply Lemma D.1 and get:

$$\mathrm{NashGap}(\pi) \le \frac{1}{(1-\gamma)^2} \cdot \max_i \mathbb{E}_{s\sim\mu_{\pi^E}}\left[\left\|\pi(\cdot|s) - \pi^E(\cdot|s)\right\|_1 + \left\|(\pi_i^*, \pi_{-i})(\cdot|s) - \pi^E(\cdot|s)\right\|_1\right]$$

Them we conclude using Lemma D.2 and our assumption:

$$\mathrm{NashGap}(\pi) \le \frac{(2n-1)\epsilon_{\mathrm{BC}} + \delta(\epsilon_{\mathrm{BC}})}{(1-\gamma)^2} \le \frac{2n\epsilon_{\mathrm{BC}} + \delta(\epsilon_{\mathrm{BC}})}{(1-\gamma)^2}$$

$\square$

## D Additional lemmas

**Lemma D.1** (Performance Difference Lemma, see e.g. Theorem IX.5 in Alatur et al. (2024))**.** *For any $\pi, \pi' \in \Pi, i \in [n]$,*

$$V_i^{\pi_i', \pi_{-i}}(\nu_0) - V_i^{\pi_i, \pi_{-i}}(\nu_0) = \frac{1}{1-\gamma}\mathbb{E}_{s,a\sim\rho_{\pi_i',\pi_{-i}}}\left[Q_i^\pi(s,a) - V_i^\pi(s)\right],$$

*where $Q_i^\pi(s,a) := r_i(s,a) + \gamma\sum_{s'\in\mathcal{S}}P(s'|s,a)V_i^\pi(s')$.*

*And more generally,*

$$V_i^{\pi'}(\nu_0) - V_i^\pi(\nu_0) = \frac{1}{1-\gamma}\mathbb{E}_{s,a\sim\rho_{\pi'}}\left[Q_i^\pi(s,a) - V_i^\pi(s)\right],$$

*Proof.* See proof of Theorem IX.5 in Alatur et al. (2024). $\square$

**Lemma D.2.** *Let $n \in \mathbb{N}$, and let $p_i, q_i \in \Delta_{m_i-1}$ be probability distributions over a discrete set of size $m_i$ isomorphic to $[m_i]$. Further, note $p = \times_{i=1}^n p_i, q = \times_{i=1}^n q_i$. Then,*

$$\sum_{j\in[m_1]\times\cdots\times[m_n]} |p(j) - q(j)| \le \sum_{i=1}^n\sum_{j=1}^{m_i}|p_i(j) - q_i(j)|$$

*Proof.* We proceed with a proof by induction.

The statement trivially holds for $n = 1$. Assume it holds for $n$, we show that it also holds for $n + 1$. For simplicity, note $S_n = \times_{i=1}^n[m_i]$ and $p^n = \times_{i=1}^n p_i, q^n = \times_{i=1}^n q_i$.

$$\sum_{j \in S_n \times [m_{n+1}]} |p(j) - q(j)| = \sum_{j_1 \in S_n} \sum_{j_2 \in [m_{n+1}]} |p^n(j_1)p_{n+1}(j_2) - q^n(j_1)q_{n+1}(j_2)|$$

$$= \sum_{j_1 \in S_n} \sum_{j_2 \in [m_{n+1}]} |p^n(j_1)p_{n+1}(j_2) - q^n(j_1)p_{n+1}(j_2)$$

$$+ q^n(j_1)p_{n+1}(j_2) - q^n(j_1)q_{n+1}(j_2)|$$

$$\leq \sum_{j_1 \in S_n} \sum_{j_2 \in [m_{n+1}]} |p^n(j_1)p_{n+1}(j_2) - q^n(j_1)p_{n+1}(j_2)|$$

$$+ \sum_{j_1 \in S_n} \sum_{j_2 \in [m_{n+1}]} |q^n(j_1)p_{n+1}(j_2) - q^n(j_1)q_{n+1}(j_2)|$$

$$= \sum_{j_1 \in S_n} |p^n(j_1) - q^n(j_1)| + \sum_{j_2 \in [m_{n+1}]} |p_{n+1}(j_2) - q_{n+1}(j_2)|$$

$$\leq \sum_{i=1}^{n} \sum_{j=1}^{m_i} |p_i(j) - q_i(j)| + \sum_{j_2 \in [m_{n+1}]} |p_{n+1}(j_2) - q_{n+1}(j_2)|$$

$$= \sum_{i=1}^{n+1} \sum_{j=1}^{m_i} |p_i(j) - q_i(j)|$$

$\square$

# E    NUMERICAL VALIDATION

To better illustrate how our theoretical insights are related to practical empirical settings, in the following, we provide the example of a game in which a tight $\delta$ function can be estimated from data.

Below, we show two main properties empirically: (1) we show that our theoretically-derived upper bound indeed holds, (2) we show the impact of entropy-regularization on the tightest delta function estimate. We would like to emphasize though that the tightness of our upper bound will ultimately depend on the precise environment and imitation setting that is considered.

For both experiments, we consider the following Tag-Game environment:

**Tag-Game**. An infinite horizon two-player zero-sum game played on a 2x3 grid. For this game, we set $\gamma = 0.8$. For each time step $t$, one of the player is the tagger, whose goal is to hit/collide with the other player who is then being chased. When the collision happens, the roles are inverted. The action spaces $\mathcal{A}_1 = \mathcal{A}_2 = \{0, 1, 2, 3\}$ represent actions corresponding to moving one step in one of the four cardinal directions. For each of the experiment, we consider results from all the initial states where players start at different corners of the grid. However, we reduce the number of initial states to two non-symmetrical states, while all the others can be recovered by symmetry of the game. The rewards are zero-sum, penalize the tagger with a constant reward, while also penalizing players from trying to escape the grid.

In this environment, we first compute the unique regularized Nash equilibrium $\pi^E$ using a Nash Value Iteration algorithm (see e.g. Perolat et al. (2015)). Then, we perturb the equilibrium at every state with different noise levels to randomly generate a set of BC policies. For each imitation policy $\pi^i$ we compute its empirical behavioral cloning error $\hat{\epsilon}_{ij}$ using $2 \cdot 10^3$ episodes to estimate the expert state distribution. We can then again apply value iteration in the induced MDPs to compute best-responses and estimate a tight delta function.

### E.1 ON THE VALIDITY OF THE UPPER BOUND (LEMMA 4)

Following Definition 6, we compute a tight $\delta$ by taking the cumulative maximum of the expected L1 norm between the expert and the best response, maximizing over the two players. Similarly, we compute the tight Nash gap upper bound as a cumulative maximum over the empirical Nash gaps.

For this experiment, we use $400$ noise levels evenly sampled from $[0, 0.4]$, and a fixed temperature of $0.1$ for computational efficiency. In Figure 5 we show how in this simple setting $\delta$ increases when $\epsilon_{\mathrm{BC}}$ increases, as well as the $\mathrm{NashGap}$.

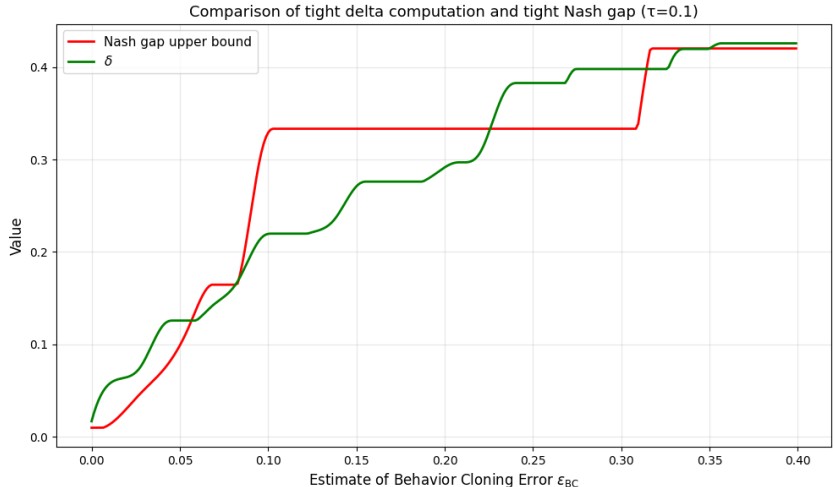

Figure 5: Evolution of the tight Nash gap upper bound and the tight $\delta$ function with the behavior cloning error. This shows how the delta function and Nash gap tend to increase together in the simple Tag-Game environment.

This plot shows that the Nash gap tends to increase together with $\delta$. Our upper bound, being greater than $\delta(\epsilon_{\mathrm{BC}})\tilde{H}^2$ with $\tilde{H} := 1/(1-\gamma)$ the effective horizon, is therefore valid.

Note that in an arbitrary toy environment, the bound cannot be required to be tight. This is expected behavior as the bound is uniform on the class of $\delta$-continuous games, thus corresponding to a worst-case scenario. In particular, the $\tilde{H}^2$ factor corresponds to the worst-case expected advantage of the best-response with respect to the expert in the newly induced MDP. A similar behavior will appear with concentrability-based bounds.

### E.2 ON THE IMPACT OF ENTROPY REGULARIZATION

We now again use Definition 6 to compute a tight estimate of $\delta(\epsilon_{\mathrm{BC}})$ for various BC errors. Related to the ideas mentioned in Section 6.3, we vary the entropy regularization factor and show its impact on $\delta$.

In Figure 6a we run the experiment by generating $250$ BC policies on a large range of $[0, 1.0]$ for the noise levels. This reveals that increasing the temperature tends to lower the $\delta(\epsilon_{\mathrm{BC}})$ when $\epsilon_{\mathrm{BC}} \in [0, 1]$ for the simple Tag-Game environment. However, we note that fixing a game $G$, this does not necessarily mean that the $\mathrm{NashGap}$ is reduced by increasing the temperature, since our upper bound Lemma 4 is not tight for $G$ in general.

The appropriate $\epsilon_{\mathrm{BC}}$ scale of importance ultimately depends on the application, the trajectory budget, expressiveness of the imitation model, as well as other factors. In the tabular setting, where $\epsilon_{\mathrm{BC}} \approx 10^{-2}$ is achievable with tens of thousands of examples (see e.g. Freihaut et al. (2025)), the behavior of regularization coefficients remains consistent, as shown in Figure 6b.

For practical work, this matters because when humans are used as experts, we often assume entropy regularization to model humans' irrationality. Having a sense of how this can also help control the exploitability of the imitation policies is therefore important.

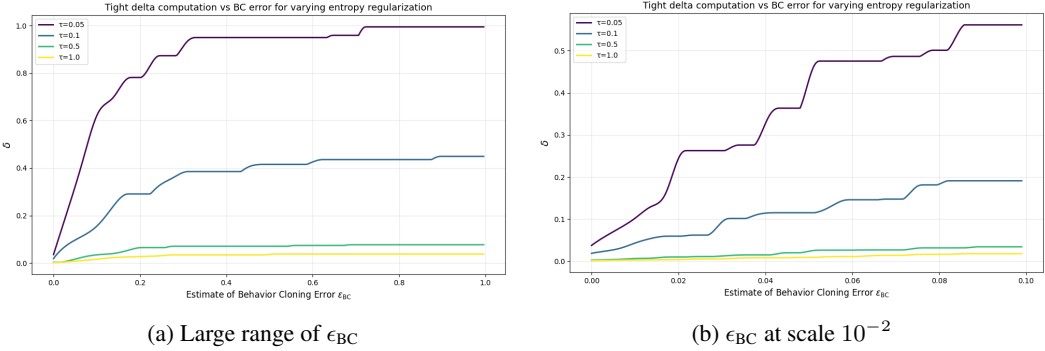

(a) Large range of $\epsilon_{\mathrm{BC}}$           (b) $\epsilon_{\mathrm{BC}}$ at scale $10^{-2}$

Figure 6: Evolution of the tight $\delta$ for different entropy temperature parameters. This shows that a greater temperature leads to a better control on $\delta$. This behavior is consistent on the full range of $\epsilon_{\mathrm{BC}}$ values.

## F  FINITE HORIZON CASE

The main content of the paper focuses on the infinite horizon case for notation simplicity. We show in this section how the results also translate to the finite horizon case. The statements are usually the same, but replacing the effective horizon $\frac{1}{1-\gamma}$ by a finite horizon $H$. We formalize this intuition below by first providing alternative definitions for the finite agent case, and then reproving our main results.

### F.1  DEFINITIONS

#### F.1.1  FINITE HORIZON MARKOV GAMES

A finite horizon $n$-player Markov Game is defined similarly to its infinite horizon equivalent with a tuple $(\mathcal{S}, \mathcal{A}, P, \{r_i\}_{i=1}^n, \nu_0, H)$. The discount factor $\gamma$ has been replaced by a finite horizon $H \in \mathbb{N}$. Further, rewards and policies of this game are now time-dependent: for all $t$, rewards are now denoted $r_i^t : \mathcal{S} \times \mathcal{A} \to [-1, 1]$ and policies become non-stationary $\pi_i^t : \mathcal{S} \to \Delta_{\mathcal{A}}$. The transition dynamics remain Markovian and $P$ is unchanged.

Because of introduced time dependence, occupancy measures are usually not generalized but denote the visitation frequencies at time t of a state and state-action pair, respectively.

$$\mu_\pi^t(s) := \mathbb{P}(s_t = s) \qquad\qquad \rho_\pi^t(s, a) := \mu_\pi^t(s)\pi^t(a|s)$$

This allows the definition of time-dependent value functions as follows

$$V_{i,t}^\pi(s) := \sum_{h=t}^{H-1} \mathbb{E}_{(s,a)\sim\rho_\pi^t}[r_i^t(s, a)] \quad \forall i \in [n],$$

For ease of notation we define $V_i^\pi(s) = V_{i,0}^\pi(s)$ for all $s \in \mathcal{S}$ and policy $\pi$. Again, the definition of value functions is extended to $V_{i,t}^\pi(\nu)$ for any distribution $\nu \in \Delta_{\mathcal{S}}$.

#### F.1.2  ASSUMPTIONS ON MATCHING ERRORS

The assumptions on errors at convergence are adapted as follows.

**BC Error:** Error from directly matching the empirical distribution of the independent individual players $\epsilon_{\mathrm{BC}} := \max_{i,t} \mathbb{E}_{s\sim\mu_{\pi^E}^t}\left[\left\|\pi_{i,t}(\cdot|s) - \pi_{i,t}^E(\cdot|s)\right\|_1\right]$

**Measure Matching Error:** Error on matching occupancy measures.

- State-only occupancy measure: $\epsilon_\mu := \max_t \left\| \mu_\pi^t - \mu_{\pi^E}^t \right\|_1$
- State-action occupancy measure: $\epsilon_\rho := \max_t \left\| \rho_\pi^t - \rho_{\pi^E}^t \right\|_1$

## F.2 PERFORMANCE DIFFERENCE LEMMA FOR FINITE HORIZON

In this section we adapt the previously stated Lemma D.1 to finite horizon Markov games. This will allow us to generalize the results of the paper in Appendix F.3.

**Lemma F.1** (Finite horizon version of Lemma D.1). *For any* $\pi, \pi' \in \Pi, i \in [n]$,

$$V_i^{\pi'_i, \pi_{-i}}(\nu_0) - V_i^{\pi_i, \pi_{-i}}(\nu_0) = \sum_{t=0}^{H-1} \mathbb{E}_{s,a \sim \rho_{(\pi'_i, \pi_{-i})}^t} \left[ Q_{i,t}^\pi(s,a) - V_{i,t}^\pi(s) \right],$$

*where* $Q_{i,t}^\pi(s,a) := r_i^t(s,a) + \sum_{s' \in \mathcal{S}} P(s'|s,a) V_{i,t+1}^\pi(s')$.

*And more generally,*

$$V_i^{\pi'}(\nu_0) - V_i^\pi(\nu_0) = \sum_{t=0}^{H-1} \mathbb{E}_{s,a \sim \rho_{(\pi')}^t} \left[ Q_{i,t}^\pi(s,a) - V_{i,t}^\pi(s) \right],$$

*Proof.* We only prove the first version. The generalization can be proven by the exact same construction.

To simplify, given a policy $\tilde{\pi}$ we overload the notations of the Q functions as follows. For every player $i \in [N]$ we denote the state-actions values of $i$ in the MDP induced by $\tilde{\pi}_{-i}$ as:

$$Q_{i,t}^{\tilde{\pi}}(s, a_i) = \mathbb{E}_{a_{-i} \sim \tilde{\pi}_{-i,t}(\cdot|s)} \left[ r_i^t(s,a) + \mathbb{E}_{s'} \left[ V_{i,t+1}^{\tilde{\pi}}(s') \right] \right], \quad 0 \le t \le H-1$$

Now, let $0 < t \le H-1$. Developing $V_{i,t}^{\pi'_i, \pi_{-i}}(\nu_t)$ for any distribution $\nu_t \in \Delta_{\mathcal{S}}$ we have

$$V_{i,t}^{\pi'_i, \pi_{-i}}(\nu_t) = V_{i,t}^{\pi'_i, \pi_{-i}}(\nu_t) - \mathbb{E}_{s \sim \nu_t, a_i \sim \pi'_{i,t}(\cdot|s)} \left[ Q_{i,t}^\pi(s,a_i) \right] + \mathbb{E}_{s \sim \nu_t, a_i \sim \pi'_{i,t}(\cdot|s)} \left[ Q_{i,t}^\pi(s,a_i) \right]$$

$$= \mathbb{E}_{s \sim \nu_t, a_i \sim \pi'_{i,t}(\cdot|s)} \left[ Q_{i,t}^{\pi'_i, \pi_{-i}}(s,a_i) \right]$$

$$\quad - \mathbb{E}_{s \sim \nu_t, a_i \sim \pi'_{i,t}(\cdot|s)} \left[ Q_{i,t}^\pi(s,a_i) \right] + \mathbb{E}_{s \sim \nu_t, a_i \sim \pi'_{i,t}(\cdot|s)} \left[ Q_{i,t}^\pi(s,a_i) \right]$$

$$= V_{i,t+1}^{\pi'_i, \pi_{-i}}(\nu_{t+1}) - V_{i,t+1}^{\pi_i, \pi_{-i}}(\nu_{t+1}) + \mathbb{E}_{s \sim \nu_t, a_i \sim \pi'_{i,t}(\cdot|s)} \left[ Q_{i,t}^\pi(s,a_i) \right]$$

where $\nu_{t+1} \in \Delta_{\mathcal{S}}$ is the distribution over the next state after following policy $(\pi'_i, \pi_{-i})$.

Subtracting $V_i^\pi(\nu_t)$ on both sides and applying the recursion we get for the initial distribution $\nu_0$,

$$V_i^{\pi'_i, \pi_{-i}}(\nu_0) - V_i^\pi(\nu_0) = \sum_{t=0}^{H-1} \mathbb{E}_{s \sim \mu_{\pi'_i, \pi_{-i}}^t} \left[ \mathbb{E}_{a_i \sim \pi'_{i,t}(\cdot|s)} \left[ Q_{i,t}^\pi(s,a_i) \right] - V_{i,t}^\pi(s) \right]$$

where we recognized $\nu_t \sim \mu_{\pi'_i, \pi_{-i}}^t$.

Rearranging the terms gives the final result

$$V_i^{\pi'_i, \pi_{-i}}(\nu_0) - V_i^{\pi_i, \pi_{-i}}(\nu_0) = \sum_{t=0}^{H-1} \mathbb{E}_{s,a \sim \rho_{(\pi'_i, \pi_{-i})}^t} \left[ Q_{i,t}^\pi(s,a) - V_{i,t}^\pi(s) \right]$$

$\square$

## F.3 RESULTS

For completeness, we reprove below our results now considering finite horizon games. The counter-examples remain largely the same, as well as the proof structures.

### F.3.1 SECTION 4

**Lemma F.2** (Finite version of Lemma 1). *There exists a game and a corresponding expert policy $\pi^E$ such that $\mathcal{S}_{\pi^E}^+ = \mathcal{S}$. Moreover, there exists a policy $\pi$ such that $\mu_{\pi^E} = \mu_\pi$ and $\mathrm{NashGap}(\pi) \geq \Omega(H)$.*

*Proof.* The example provided in the main text is also valid to prove this lemma, by assuming an arbitrary horizon H. □

**Theorem F.1** (Finite horizon version of Theorem 2). *There exists a Markov Game with expert policy $\pi^E$ and a learner policy $\pi$ such that even if $\rho_{\pi^E}^t = \rho_\pi^t$ for all $0 \leq t \leq H - 1$, the Nash gap scales linearly with the horizon; i.e., $\mathrm{NashGap}(\pi) \geq \Omega(H)$.*

*Proof.* The proof is exactly the same as in Appendix B.3, except the state space which is finite (therefore still countable). □

### F.3.2 SECTION 6

**Lemma F.3** (Finite horizon version of Lemma 2). *Let $\mathcal{C}$ be the class of games with consistent bounds. This class is $\delta$-continuous only for trivial $\delta$ such that $\delta(\epsilon) = 2$ for all $\epsilon > 0$.*

*Proof.* A similar proof to the one provided in the main text is valid for the finite horizon case. It suffices to adapt the $k$ for the finite horizon, ensuring we keep the property $\mu_{\pi^E}(s_{\exp}) \leq \epsilon/2$. □

**Lemma F.4** (Finite horizon version of Lemma 3). *Suppose $\pi^E$ is a (weak) Dominant Strategy Equilibrium. Then, any learned policy $\pi$ with BC error $\epsilon_{BC}$ satisfies $\mathrm{NashGap}(\pi) \leq 2n\epsilon_{BC}H^2$.*

*Proof.* The Dominant Strategy Equilibrium assumption gives:

$$\mathrm{NashGap}(\pi) = \max_{i,\pi_i^*} \left[ V_i^{\pi_i^*, \pi^{-i}}(\nu_0) - V_i^{\pi_i, \pi^{-i}}(\nu_0) \right]$$

$$= \max_i \left[ V_i^{\pi_i^E, \pi^{-i}}(\nu_0) - V_i^{\pi^E}(\nu_0) + V_i^{\pi^E}(\nu_0) - V_i^{\pi_i, \pi^{-i}}(\nu_0) \right]$$

Fixing $i$, we apply the Performance Difference Lemma (Lemma F.1) to get:

$$V_i^{\pi^E}(\nu_0) - V_i^{\pi_i, \pi^{-i}}(\nu_0) = \sum_{t=0}^{H-1} \mathbb{E}_{s,a \sim \rho_{\pi^E}^t} \left[ Q_{i,t}^\pi(s,a) - V_{i,t}^\pi(s) \right]$$

$$= \sum_{t=0}^{H-1} \mathbb{E}_{s \sim \mu_{\pi^E}^t} \left[ \sum_a Q_{i,t}^\pi(s,a) \left( \pi_t^E(a|s) - \pi_t(a|s) \right) \right]$$

$$\leq H \sum_{t=0}^{H-1} \mathbb{E}_{s \sim \mu_{\pi^E}^t} \left[ \left\| \pi_t^E(\cdot|s) - \pi_t(\cdot|s) \right\|_1 \right]$$

$$\leq H^2 \cdot \max_{0 \leq t \leq H-1} \mathbb{E}_{s \sim \mu_{\pi^E}^t} \left[ \left\| \pi_t^E(\cdot|s) - \pi_t(\cdot|s) \right\|_1 \right]$$

where the Q-functions are defined as in Lemma F.1.

Similarly, applying the PDL in the reverse order,

$$V_i^{\pi_i^E, \pi^{-i}}(\nu_0) - V_i^{\pi^E}(\nu_0) = \sum_{t=0}^{H-1} \mathbb{E}_{s,a \sim \rho_{\pi^E}^t} \left[ V_{i,t}^{\pi_i^E, \pi^{-i}}(s) - Q_{i,t}^{\pi_i^E, \pi^{-i}}(s,a) \right]$$

$$= \sum_{t=0}^{H-1} \mathbb{E}_{s \sim \mu_{\pi^E}^t} \left[ \sum_a Q_{i,t}^{\pi_i^E, \pi^{-i}}(s,a) \left( \pi_{-i,t}(a|s) - \pi_{-i,t}^E(a|s) \right) \pi_{i,t}^E(a_i|s) \right]$$

$$\leq H \sum_{t=0}^{H-1} \mathbb{E}_{s \sim \mu_{\pi^E}^t} \left[ \left\| \pi_{-i,t}(\cdot|s) - \pi_{-i,t}^E(\cdot|s) \right\|_1 \right]$$

$$\leq H^2 \cdot \max_{0 \leq t \leq H-1} \mathbb{E}_{s \sim \mu_{\pi^E}^t} \left[ \left\| \pi_{-i,t}(\cdot|s) - \pi_{-i,t}^E(\cdot|s) \right\|_1 \right]$$

We conclude using Lemma D.2.

$$\text{NashGap}(\pi) \leq (2n - 1)\epsilon_{\text{BC}}H^2 \leq 2n\epsilon_{\text{BC}}H^2$$

$\square$

**Lemma F.5** (Finite horizon version of Lemma 4). *Let $\pi$ be the learned policy, and assume for all $i \in [n]$ and $0 \leq t \leq H - 1$ that $\mathbb{E}_{s \sim \mu_{\pi^E}^t} \left[ \left\| \pi_{i,t}^*(\cdot|s) - \pi_{i,t}^E(\cdot|s) \right\|_1 \right] \leq \mathcal{O}(\delta(\epsilon_{BC}))$ for some function $\delta$, where $\pi_i^* \in \text{BR}_i(\pi)$. Then, $\text{NashGap}(\pi) \leq (2n\epsilon_{BC} + \delta(\epsilon_{BC})) H^2$.*

*Proof.* Using similar arguments, applying Lemma F.1 and Lemma D.2:

$$\text{NashGap}(\pi) \leq \max_i \left( \max_{0 \leq t \leq H-1} \mathbb{E}_{s \sim \mu_{\pi^E}^t} \left[ \left\| \pi_t^E(\cdot|s) - \pi_t(\cdot|s) \right\|_1 \right] \right.$$

$$\left. + \max_{0 \leq t \leq H-1} \mathbb{E}_{s \sim \mu_{\pi^E}^t} \left[ \left\| \pi_t^E(\cdot|s) - (\pi_{i,t}^*, \pi_{-i,t})(\cdot|s) \right\|_1 \right] \right) H^2$$

$$\leq \left( (2n - 1)\epsilon_{\text{BC}} + \delta(\epsilon_{\text{BC}}) \right) H^2$$

$$\leq \left( 2n\epsilon_{\text{BC}} + \delta(\epsilon_{\text{BC}}) \right) H^2$$

$\square$

