# OpenReview forum: "Matching Multiple Experts: On the Exploitability of Multi-Agent Imitation Learning"
_ICLR.cc/2026/Conference — ICLR 2026 Poster_

### Official Review · Reviewer_znH1 · 2025-10-31

**Soundness:** 3
**Presentation:** 3
**Contribution:** 2
**Rating:** 6
**Confidence:** 4

**Summary:**

This paper studies imitation learning in Markov games. It first demonstrates that several previously used metrics, such as BC error or measure matching error, fail to provide theoretical bounds on the performance of learned equilibria. The paper then establishes that obtaining exact lower bounds for the Nash gap is PPAD-hard. Finally, the paper proposes a new notion of best-response continuity and provides theoretical guarantees on the Nash gap under the new condition.

**Strengths:**

- Overall the paper is well-written, the question studied in this paper is clearly stated and most proofs are easy to follow.
- The paper provides constructive examples where BC error and measure matching error fail to capture the performance of learned equilibria.
- Although the paper is not mathematically heavy, I believe the result carries some significance to the game theory community.

**Weaknesses:**

- One issue with the definition of the tight Nash gap lower bound is that $\mathcal{M}_{\epsilon}(\pi^{E})$ is not a convex set due to the equality constraint. Given this non-convex nature, it is not surprising that computing this bound leads to some hardness results (maybe even NP-hardness). While it is understandable to impose such equality constraint, the authors could consider more natural way to capture this gap.

- To obtain a meaningful result from Lemma 4, one would expect $\delta(\epsilon)$ to be a polynomial function of $\epsilon$. However, this essentially implies that when $\epsilon = 0$, $\pi_i^{E}$ is the unique best response policy given $\pi_{-i}^{E}$. This notion appears quite restrictive as it is well-known that multiple best-response strategies may exist against a fixed policy.

- No numerical experiments are provided.

**Questions:**

- I find it hard to believe that Theorem 3 holds for any $\epsilon_{\rho} > 0.$ It is well-known that computing $\epsilon$-Nash for some constant $\epsilon$ is easy (see [1] for example). What is the connection between $\epsilon_{\rho}$ and the approximation error $\epsilon$ in your reduction?

----
[1] Constantinos Daskalakis, Aranyak Mehta, and Christos H. Papadimitriou. A note on approximate nash equilibria. Theor. Comput. Sci., 410(17):1581–1588, 2009.

---

> ### Author Response · Authors · 2025-11-21
>
> Thank you for helping us improve the current work. We provide below some clarifications and answers to the points that you raised.
>
> **On the weaknesses**
>
> > One issue with the definition of the tight Nash gap lower bound is that  is not a convex set due to the equality constraint. Given this non-convex nature, it is not surprising that computing this bound leads to some hardness results (maybe even NP-hardness). While it is understandable to impose such equality constraint, the authors could consider more natural way to capture this gap.
>
> We note that given our definition of $m_{\rho}$, the following tight inequality holds for any game $G$ and corresponding expert $\pi^E$:
> $$
> \operatorname{NashGap}(\pi, \pi^E) \geq m\_{\rho}(G, \epsilon\_{\rho}), \quad \forall \pi \in \mathcal{M}\_{\epsilon\_{\rho}}(\pi^E)
> $$ This motivates our definition. We provide more details below in the answer to the question.
>
> > To obtain a meaningful result from Lemma 4, one would expect $\delta(\epsilon)$ to be a polynomial function of $\epsilon$. However, this essentially implies that when $\epsilon=0$, $\pi^E_{i}$ is the unique best response policy given $\pi^E_{-i}$. This notion appears quite restrictive as it is well-known that multiple best-response strategies may exist against a fixed policy.
>
> Indeed, it would not be possible to have $\delta(0) = 0$ in the case of multiple best-responses to a fix strategy $\pi^E_{-i}$. Note that this case is not directly applied from having multiple Nash equilibria in a game, but is implied from having at least one mixed equilibrium.
>
> > No numerical experiments are provided.
>
> Please see our general answer to all reviewers.
>
> **Question**
> > I find it hard to believe that Theorem 3 holds for any $\epsilon_{\rho} > 0$. It is well-known that computing $\epsilon$-Nash for some constant $\epsilon$ is easy (see [1] for example). What is the connection between $\epsilon_{\rho}$ and the approximation error $\epsilon$ in your reduction?
>
> Thank you for raising this question. We agree that the theorem doesn't hold for a fixed $\epsilon$. For example, it is easy to see that extreme values of $\epsilon$ can lead to trivial Nash gaps. We have clarified the statement of our result in a new revision of the paper to better convey this.
>
> We would like to clarify that what is PPAD-hard is the problem of constructing a solver that for any $\epsilon_{\rho}$ and any game $G$ outputs the tight Nash gap $m\_{\rho}(G, \epsilon\_{\rho})$ in polynomial time. One might perform imitation learning and want to track an optimistic estimate on the exploitability (Nash gap) of the learned solution. Theorem 3 is saying that we cannot a priori construct an efficient algorithm for this problem.

---

> > ### Comment · Reviewer_znH1 · 2025-11-28
> >
> > I thank the authors for their rebuttal, I would like to keep my original score.

---

### Official Review · Reviewer_itJa · 2025-10-31

**Soundness:** 2
**Presentation:** 4
**Contribution:** 3
**Rating:** 6
**Confidence:** 4

**Summary:**

This theoretical paper studies multi-agent imitation learning with the goal of recovering a Nash equilibrium from demonstrations generated by an expert equilibrium. Exploitability is defined as the distance of the learned policy to a Nash equilibrium, and the authors derive both consistent bounds, which vanish with imitation error, and tractable bounds, which can be efficiently computed. They show that even perfectly matching expert occupancy measures can yield exploitable policies, proving the impossibility of deriving consistent Nash equilibria or tight tractable exploitability lower bounds in general Markov Games. To address this, they introduce the notion of best-response continuity, which allows constructing tractable upper bounds and provides conditions under which consistent and tractable Nash gap bounds can be achieved.

**Strengths:**

* Excellent exposition of the concepts, related work, and background. It is difficult to do based on the sheer quantity of concepts that the work introduces and uses, which makes it all the more impressive and appreciated. Sec. 3 is well balanced with just enough depth to grasp the problem.
* The proof outline of Th. 3 is appreciated and very useful.
* The paper presents its results with an extremely high level of polish.

**Weaknesses:**

* The finding that deriving exploitability upper bounds ultimately depnds on characterizing delta (and on the conditions that make delta consistent and tractable) is an interesting and valuable contribution. Still, the paper could better connect this result to more practical considerations. It would help to illustrate how specific techniques or settings might influence/shape delta in applied contexts. The brief mentions of “promoting exploration” and “penalizing risk aversion” (last lines of Sec. 6) move in that direction by linking delta to concrete mechanisms, but expanding on these examples could make the results feel more actionable. In particular, clarifying how delta might be operationalized or controlled through standard control or learning methods (practical “knobs”) would strengthen the paper, since the central result of the paper depends entirely on delta behaving well enough.

**Questions:**

* Having tractable bounds is nice, so why not craft a scenario where the last bound involving delta is tractable and show empirically how tight or simply how informative it is, in a simple toy application?
* I believe the paper provides important and sound theoretical contributions, and that the exposition is such that even non-seasoned researchers can learn valuable concepts from it. That being said, the ultimate bounds derived in the work provide limited intuitive value. Adding practical “knobs” that give insights as to how to control such bounds could go a long way in that direction.

Style, typos, suggestions:
* [minor] L085-086: "MA-ILR" -> "MA-IL" or rather "MA-IRL"?

---

> ### Author Response · Authors · 2025-11-21
>
> We thank you for all the positive comments related to the presentation of the paper. We also appreciate the constructive feedback on the practical takeaways from our main results. Please find below our answers.
>
> > Having tractable bounds is nice, so why not craft a scenario where the last bound involving delta is tractable and show empirically how tight or simply how informative it is, in a simple toy application?
>
> The tightness of our upper bound will ultimately depend on the precise environment and imitation setting. Note that our bound not making assumptions beyond $\delta$-continuity, it corresponds to a worst-case analysis of the Nash gap. However, as noted in our answer to all reviewers, we now provide empirical validation in **Appendix E** in a revision of the paper. Deriving theoretical bounds for good delta functions in certain classes of Markov games remains an interesting question for future research.
>
> As explained in **Appendix E**, the bound cannot be required to be tight in a toy environment. This is the result of having a bound that is uniform on the class of $\delta$-continuous games, corresponding to a worst-case scenario.
>
> > I believe the paper provides important and sound theoretical contributions, and that the exposition is such that even non-seasoned researchers can learn valuable concepts from it. That being said, the ultimate bounds derived in the work provide limited intuitive value. Adding practical “knobs” that give insights as to how to control such bounds could go a long way in that direction.
>
> We state here intuition for both the bound and the practical "knobs".
>
> **On the bound itself**
>
> A perhaps more intuitive manner of seeing our derived bound is to see that by the Performance Difference Lemma [1], the Nash gap boils down to an expectation of some advantage function $\operatorname{Adv}^{\tilde{\pi}}\_i$ as: $\operatorname{NashGap}(\pi, \pi^E) = \tilde{H} \cdot \mathbb{E}\_{\tilde{\pi}}[\operatorname{Adv}\_i^{\tilde{\pi}}(s)]$, for the effective horizon $\tilde{H}$ and some $i, \tilde{\pi}$. Decomposing this, we get
> $$\operatorname{Adv}\_i^{\tilde{\pi}}(s) = \langle Q\_i^{\tilde{\pi}}(s, \cdot), \tilde{\pi}(\cdot | s) - \pi^E(\cdot | s)  \rangle \leq \frac{1}{2} \tilde{H} \operatorname{TV}(\tilde{\pi}(\cdot | s), \pi^E(\cdot | s))$$ hence $\operatorname{NashGap}(\pi, \pi^E) \leq \frac{1}{2} \tilde{H}^2 \cdot \mathbb{E}_{\tilde{\pi}}[\operatorname{TV}(\pi(\cdot | s), \pi^E(\cdot | s))]$.
>
> By a change of measure introducing a concentrability coefficient $C_{\tilde{\pi}, \pi^E}$ [2, 3], we get a gap of order $\mathcal{O}(C\_{\tilde{\pi}, \pi^E} \tilde{H}^2 \epsilon\_{\text{BC}})$. Intuitively, such ratio of occupancy measures should not be necessary in general, as from Hölder's inequality we get
> $$
> \mathbb{E}\_{\tilde{\pi}}[\operatorname{TV}(\pi(\cdot | s), \pi^E(\cdot | s))] \leq \mathbb{E}\_{\pi^E}[\operatorname{TV}(\pi(\cdot | s), \pi^E(\cdot | s))] + 2 \lVert \mu\_{\tilde{\pi}} - \mu\_{\pi^E} \rVert\_1
> $$ That would lead us to a Nash gap of order $\mathcal{O}(\tilde{H}^3 (\delta(\epsilon_{\text{BC}}) + n\epsilon_{\text{BC}}))$, but without trivial bounds whenever the best-response explores states not visited by the expert. From our finer analysis, we get the quadratic dependence on $\tilde{H}$.
>
> **On practical "knobs"**
>
> As mentioned, we included some ideas in the paper that could help practitioners. Regularization, in particular, is of great importance due to its ability to correct the idealized assumption of a perfect Nash equilibrium. To support our arguments, **Figure 6** in the new **Appendix E** of the paper provides results related to the impact of entropy regularization.
>
> As noted there, for the experiment considered, a higher temperature leads to a better control on $\delta$. This behavior is consistent on the full range of $\epsilon_{\text{BC}}$ values. Because practitioners might not be interested in the full $[0, 1]$ range for $\epsilon_{\text{BC}}$, we also zoom into the range where $\epsilon_{\text{BC}} \approx 10^{-2}$, corresponding to collecting tens of thousands of samples in the tabular setting. More information are provided in **Appendix E**.
>
> Finding more of these "knobs" could be of interest for future work.
>
> > [minor] L085-086: "MA-ILR" -> "MA-IL" or rather "MA-IRL"?
>
> Thank you for pointing this out, this typo has been fixed in a new revision of the paper.
>
> References:
>
> [1] Pragnya Alatur, Anas Barakat, and Niao He. "Independent policy mirror descent for markov potential
> games: Scaling to large number of players". *Conference on Decision and
> Control (CDC)*, 2024.
>
> [2] Freihaut, Till and Viano, Luca and Cevher, Volkan and Geist, Matthieu and Ramponi, Giorgia. "Learning Equilibria from Data: Provably Efficient Multi-Agent Imitation Learning". *Conference on Neural Information Processing Systems*, 2025.
>
> [3] Cui, Qiwen and Du, Simon S. "When are offline two-player zero-sum Markov games solvable?". *Advances in Neural Information Processing Systems*, 2022.

---

### Official Review · Reviewer_ngLP · 2025-10-31

**Soundness:** 3
**Presentation:** 2
**Contribution:** 3
**Rating:** 4
**Confidence:** 3

**Summary:**

This paper investigates the theoretical limits of learning Nash-equilibrium-like policies from expert demonstrations in multi-agent imitation learning (MA-IL), especially in offline settings.  The authors first prove impossibility results, showing that even exact occupancy measure matching may lead to highly exploitable policies in general Markov games.  They further establish the PPAD-hardness of computing tight lower bounds on exploitability, linking the challenge to classical game-theoretic complexity.  Finally, the paper introduces a novel concept of best-response δ-continuity, allowing derivation of tractable and consistent upper bounds on the Nash gap—particularly under dominant strategy equilibria (DSE). The results unify several prior observations and provide theoretical guidance for designing robust MA-IL algorithms.

**Strengths:**

- The paper presents a strong theoretical contribution, providing clear impossibility and hardness results that formalize the limits of multi-agent imitation learning (MA-IL).
- The proposed δ-continuity framework is a novel and interesting concept that bridges equilibrium stability and policy learning theory.
- The PPAD-hardness argument is rigorous and well-motivated.
- The proofs are detailed and supported by illustrative examples (though I did not check all the details).
- The work clarifies when common imitation learning methods (such as BC and GAIL) can or cannot recover equilibria, which is of both theoretical and practi

**Weaknesses:**

- My major concern is that, although the paper is theoretically strong, it lacks empirical validation to demonstrate how the theoretical insights (e.g., δ-continuity effects and dominance structures) manifest in practical game settings.
- The theory assumes access to known imitation errors (e.g., Equation 2), but it does not discuss how these quantities can be estimated or bounded in real applications.
- The assumptions of dominant strategy equilibria (DSE) or δ-continuity, while mathematically elegant, are quite restrictive and may not hold in general-sum or more complex multi-agen

**Questions:**

Please address the concerns in Weakness, especially that related to empirical validation

**Details Of Ethics Concerns:**

- There are no major ethical concerns associated with this work. The paper is purely theoretical and does not involve human subjects, sensitive data, or potentially harmful application

---

> ### Author Response · Authors · 2025-11-21
>
> Thank you for your feedback and the useful comments. Please find below our answers to the questions and weaknesses pointed out.
>
> > [The paper] lacks empirical validation to demonstrate how the theoretical insights manifest in practical game settings.
>
> Please see our answer to all reviewers.
>
> > The theory assumes access to known imitation errors, but it does not discuss how these quantities can be estimated or bounded in real applications.
>
> The goal of our work is to derive exploitability bounds that can *generalize to a various sets of training and data collection procedures*, i.e., we make this assumption of known imitation errors to remain agnostic to the particular problem setup. We thank you, however, for the opportunity to give pointers on how to measure or estimate this error in some representative setups.
>
> - **Tabular setting.** In the tabular case, the data is often assumed to contain $N$ trajectories sampled from $\\pi^E$ (see e.g. Protocol 1 in [1]). As a result, the imitation policy $\\pi$ is naturally estimated as the empirical policy $\\pi(a|s) = N_{a, s}/N_s$, where $N_s$ (resp. $N_{a, s}$) denotes the number of times the state $s$ (resp. the state-action pair $(s, a)$) appears in the dataset. In that case, we can show that the following PAC bound holds: with probability greater than $1-\\delta$, $\\epsilon_{\\text{BC}} \\leq \\epsilon$ given that
> $N \\geq \\frac{16A_{max}|S|\\log^2(2n|S|/\\delta)}{\\epsilon^2},$
> where $A\_{max} = \\max_i |\\mathcal{A}\_i|$, $\\delta \\in (0, 1)$. This follows from a similar proof to Lemma C.1 in [2].
> Therefore, we are ensured that with high probability the BC error is "small" given that we collected a sufficient amount of data. Applying **Corollary 3** in our paper with a DSE expert, we then get that the imitation policy is an $\\epsilon$-Nash equilibrium with probability greater than $1-\\delta$ if
> $N \\geq \\frac{64n^2A_{max}|S|\\log^2(2n|S|/\\delta)}{\\epsilon^2(1-\\gamma)^4},$
> for any $\\delta \\in (0, 1)$.
> The same logic applies when $N$ is fixed, for example when using pre-collected data. In this case we get
> $\\epsilon_{\\text{BC}} \\leq 4\\sqrt{\\frac{A\_{max}|S|\\log^2(2n|S|/\\delta)}{N}},$
> and we can apply either **Lemma 3** or **Lemma 4** providing Nash gap guarantees for the empirical estimate $\\pi$.
>
> - **Deep learning setting.** When learning in the deep setting, the optimization loss (e.g. equation $(16)$ in GAIL [3] or negative log-likelihood for BC) is typically minimized with the imitation errors. Precise measures of the imitation errors would however be either hard to derive or unpractical.
> In this scenario, we suggest using empirical estimates from the data as a first approximation. We would estimate the BC error using a random batch $B$ of size $N$ from the dataset as $\\hat{\\epsilon}\_{\\text{BC}} = \\max_i \\sum_{s}\\frac{N\_s}{N}\\sum\_{a_i} |\\pi\_i(a_i | s) - N\_{a_i, s}/N\_s|$, extending the above notations to $B$. Estimating measure matching errors (**Equations (3) and (4)**) can be done similarly from a random batch and simulating the imitation policy in the environment. Other alternatives also exist. For the case of the loss of equation $(15)$ in GAIL [3], one can invert a lower bound on the Jensen-Shannon divergence between occupancy measures (Proposition 3.4 in [4]), and obtain an upper bound on the imitation errors as a function of the loss value.
> The most appropriate method highly depends on the loss function and the dataset.
>
> > The assumptions of DSE or δ-continuity, while mathematically elegant, are quite restrictive and may not hold in general-sum or more complex multi-agen
>
> Indeed, we agree that the dominant-strategy structure is restrictive, and this is what makes our study important. Our analysis shows that when a mild dominance property holds, one can obtain clean and computable exploitability bounds. However, once this structure is removed, MA-IL becomes genuinely hard: the Nash gap can behave erratically under behavioral cloning, and exploitability guarantees can deteriorate far beyond their single-agent counterparts. This fact highlights how dominance impacts the stability of MA-IL, and is an important contribution of our work.
>
> References:
>
> [1] Viano, Luca and Skoulakis, Stratis and Cevher, Volkan. "Imitation learning in discounted linear mdps without exploration assumptions". *International Conference on Machine Learning*. PMLR, 2024.
>
> [2] Freihaut, Till and Viano, Luca and Cevher, Volkan and Geist, Matthieu and Ramponi, Giorgia. "Learning Equilibria from Data: Provably Efficient Multi-Agent Imitation Learning". *Conference on Neural Information Processing Systems*, 2025.
>
> [3] Ho, Jonathan and Ermon, Stefano. "Generative adversarial imitation learning". *Advances in neural information processing systems*, 2016.
>
> [4] Corander, Jukka and Remes, Ulpu and Koski, Timo. "On the Jensen-Shannon divergence and the variation distance for categorical probability distributions". *Kybernetika*, Volume 57 pages 879-907, 2021.

---

### Author Response · Authors · 2025-11-21
**General answer to all reviewers about numerical experiments**

Thank you all for your valuable reviews.

Multiple comments were made about numerical experiments, and we now added numerical validation in **Appendix E** of a new revision of the paper.

Specifically, for a small toy environment described there, **Figure 5** demonstrates the validity of our bound and **Figure 6** shows the impact of entropy regularization on the tightest $\delta$ estimate one can compute from the data.

As outlined in the paper, this has both practical and theoretical implications, especially for entropy regularization which is highly used in practice.

---

### Comment · Area_Chair_9Ek6 · 2025-11-27
**Reminder: Please Discuss**

All Reviewers,

Thank you for your time. As the rebuttal has been available for a while, please engage in discussions with the authors and with one another. There are only a few days left before December 3.

Best,
Area Chair

---

### Meta-Review · Area_Chair_W5Ym · 2026-01-04

**Summary:**

Across the reviews, the paper is consistently recognized as a theoretical contribution to multi-agent imitation learning (MA-IL). Reviewers agree that the paper formalizes important limitations of MA-IL by showing that common metrics such as behavioral cloning error and occupancy measure matching do not guarantee low exploitability, and by proving PPAD-hardness of computing tight Nash-gap lower bounds in general Markov games. The introduction of best-response δ-continuity is viewed as a meaningful conceptual contribution that enables the derivation of tractable upper bounds on exploitability under additional assumptions (e.g., dominant strategy equilibria). The exposition and proof structure are repeatedly praised, with reviewers highlighting clarity, polish, and accessibility despite the technical nature of the results. The main concerns informing the decision focus on the lack of empirical validation and the limited practical interpretability of the theoretical bounds. Multiple reviewers explicitly noted that the original submission contained no numerical experiments and requested illustrative or toy examples to demonstrate how the bounds behave and how quantities such as δ or entropy regularization affect exploitability in practice. In response, the authors added numerical experiments in Appendix E, including a toy environment validating the derived bound and an analysis of the effect of entropy regularization on the tightest computable exploitability estimate.

**Reviewer Concerns:**

Concerns addressed by the rebuttal and revision:
1. Absence of numerical experiments (ngLP, itJa, znH1): The authors added toy numerical experiments in Appendix E. Figure 5 illustrates the validity of the bound in a small environment, and Figure 6 analyzes how entropy regularization influences the tightest exploitability estimate computable from data.
2. Estimability of imitation errors (ngLP): The authors provided detailed explanations for how imitation errors can be estimated or bounded in tabular settings (via empirical policies and PAC-style arguments) and suggested empirical estimation strategies in deep-learning settings, including batch-based estimates and loss-based bounds.
3. Clarification of theoretical statements (znH1): The authors clarified the scope of the PPAD-hardness result, explicitly noting that it does not apply to fixed-ε approximate Nash equilibria, and revised the paper to better convey this distinction.
4. Interpretation of δ and practical “knobs” (itJa): The rebuttal expands on how regularization, particularly entropy regularization, can influence δ, and connects this discussion directly to the new toy experiments in Appendix E.
5. Presentation issues and typos (itJa): Minor notation and typographical issues were acknowledged and corrected in the revised version.

Concerns partially addressed or still outstanding:
1. Limited empirical scope: Although the added experiments provide illustrative validation, they are restricted to toy environments and do not demonstrate how the theoretical insights translate to realistic or applied MA-IL settings.
2. Restrictive assumptions: Reviewers note that the reliance on dominant strategy equilibria or δ-continuity is mathematically elegant but restrictive; while the authors argue this is a key insight of the work, it limits direct applicability.

**Reviewer Scores:**

Reviewer ngLP: Likely unchanged. While the addition of toy experiments addresses the explicit request for empirical validation, the reviewer’s concern about how theory manifests in practical settings is only partially resolved.

Reviewer itJa: Likely unchanged. This reviewer feels positive about the theoretical contribution and presentation, but views the bounds as offering limited intuitive or practical guidance.

Reviewer znH1: Explicitly unchanged. The reviewer acknowledged the clarifications and stated they would keep their original score.

---

### Decision · Program_Chairs · 2026-01-26

Accept (Poster)